# High-sensitivity in situ capture of endogenous RNA-protein interactions in fixed cells and primary tissues

Qishan Liang [1,2,8], Tao Yu [2,3,4,5,8], Eric Kofman[3,4,5,6], Pratibha Jagannatha [3,4,5,6], Kevin Rhine[3,4,5], Brian A. Yee [2,3,4,5], Kevin D. Corbett[2,3,7] ✉ & Gene W. Yeo [2,3,4,5,6] ✉

RNA-binding proteins (RBPs) have pivotal functions in RNA metabolism, but current methods are limited in retrieving RBP-RNA interactions within endogenous biological contexts. Here, we develop INSCRIBE (IN situ Sensitive Capture of RNA-protein Interactions in Biological Environments), circumventing the challenges through in situ RNA labeling by precisely directing a purified APOBEC1-nanobody fusion to the RBP of interest. This method enables highly specific RNA-binding site identification across a diverse range of fixed biological samples such as HEK293T cells and mouse brain tissue and accurately identifies the canonical binding motifs of RBFOX2 (UGCAUG) and TDP-43 (UGUGUG) in native cellular environments. Applicable to any RBP with available primary antibodies, INSCRIBE enables sensitive capture of RBP-RNA interactions from ultra-low input equivalent to ~5 cells. The robust, versatile, and sensitive INSCRIBE workflow is particularly beneficial for precious tissues such as clinical samples, empowering the exploration of genuine RBP-RNA interactions in RNA-related disease contexts.

As pivotal players in the life cycle of mRNAs and other functional RNAs, RNA-binding proteins (RBPs) are essential for RNA modification, splicing, stability, localization, translation, and many other post-transcriptional regulatory events[1–3]. Dysregulation of RBPs can give rise to many diseases including neurodegenerative disorders and cancer[4–6] due to disrupted RBP-RNA interactions and consequent improper RNA processing. Therefore, mapping RBP-RNA interaction profiles across the transcriptome is crucial to unravel RBP functions and understand RNA-related diseases.

Extensive efforts have been undertaken to characterize RBP-RNA interactions at the molecular and transcriptomic levels. The most widely-used RBP-RNA interaction profiling approaches are immunoprecipitation (IP)-based technologies such as CLIP-seq (Cross-Linking and Immunoprecipitation followed by sequencing)[7–10], which have significantly advanced our understanding of RBP-RNA interactions in various biological contexts. However, these methods are laborious and require substantial starting materials due to the loss of RNA during immunoprecipitation and low cross-linking efficiency[11]. These drawbacks preclude the capture of RBP-RNA interactions in precious primary tissue samples and clinical samples. Additionally, the RNA digestion in CLIP-seq protocols obfuscates the distinction between RNA isoforms, which may have different regulatory functions in RNA metabolism.

[1]Department of Chemistry and Biochemistry, University of California San Diego, La Jolla, CA, USA. [2]Center for RNA Technologies and Therapeutics, University of California San Diego, La Jolla, CA, USA. [3]Department of Cellular and Molecular Medicine, University of California San Diego, La Jolla, CA, USA. [4]Sanford Stem Cell Institute and Stem Cell Program, University of California San Diego, La Jolla, CA, USA. [5]Institute for Genomic Medicine, University of California San Diego, La Jolla, CA, USA. [6]Bioinformatics and Systems Biology Program, University of California San Diego, La Jolla, CA, USA. [7]Department of Molecular Biology, University of California San Diego, La Jolla, CA, USA. [8]These authors contributed equally: Qishan Liang, Tao Yu. ✉e-mail: kcorbett@ucsd.edu; gene.w.yeo@gmail.com

Recently developed tools such as STAMP (Surveying Targets by APOBEC-Mediated Profiling)[12] and TRIBE (Targets of RNA-binding proteins Identified By Editing)[13] address the limitations of IP-based profiling methods by using RNA modifying enzymes to label RNA bases in proximity to an RBP of interest in living cells. Without IP enrichment or RNA digestion steps, these tools can identify RNA isoforms with minimal input materials, including at the single-cell level[12]. However, both methods rely on overexpression of an exogenous RBP-enzyme fusion protein in living cells, which could lead to artifacts in the RBP's localization, functionality, and RNA-binding characteristics. While informative, the resulting RBP-RNA interaction profiles may not accurately reflect true RBP-RNA interaction profiles in endogenous biological environments. In addition, non-engineerable tissues such as clinical biopsy samples are not amenable to interrogation via these methods.

Here we present INSCRIBE (IN situ Sensitive Capture of RNA-protein Interactions in Biological Environments), a unique and versatile solution that combines the advantages of CLIP and STAMP while circumventing their respective disadvantages, preserving an authentic RBP-RNA interaction map in undisrupted biological contexts. This technology harnesses a purified recombinant protein APOBEC1-nanobody that is precisely directed to the RBP of interest through the nanobody-primary antibody recognition. Through in situ RNA C-to-U labeling by the cytosine deaminase APOBEC1 with a simple workflow, INSCRIBE identifies transcriptomic RBP-RNA interactions in unmodified cells and primary tissues fixed by either methanol or formaldehyde, without requiring any prior plasmid constructions and transformations into cells (Fig. 1a). Coupled with an established computational pipeline that maps C-to-U conversion in the transcriptome (Fig. 1b), INSCRIBE enables robust identification of endogenous RBP binding sites in fixed HEK293T cells and mouse brain tissue slices.

INSCRIBE is highly specific as it readily enriches the consensus binding motifs of two RBPs (RBFOX2 and TDP-43) and agrees with orthogonal methods such as eCLIP. The technique is compatible with PacBio cDNA long-read sequencing, enabling the distinction between different RNA isoform substrates. One of the advantages of INSCRIBE is its versatility: With a single recombinant enzyme-nanobody fusion, INSCRIBE can profile diverse RBPs with commercially available primary antibodies bearing the same IgG for nanobody recognition. Another great advantage of INSCRIBE lies in its high sensitivity and low-input material requirement. While routine INSCRIBE protocol uses ~150,000 cells or a slice of primary tissue, it can capture transcriptome-wide endogenous RBP-RNA interactions with as little as 100 pg of RNA, equivalent to ~5 cells, compared to the 5–10 million cells used in a typical eCLIP experiment. Additionally, the workflow is much less laborious and time-consuming, as the INSCRIBE experiment only takes 20–24 h prior to RNA-seq library preparation, compared to ~4.5 days in the eCLIP workflow. The robust, versatile, sensitive, and convenient INSCRIBE method is particularly beneficial when handling preserved valuable samples, such as clinical samples and patient-derived stem cells. We envision INSCRIBE's broad application to bridge the critical gap in unveiling RBP roles in RNA-linked diseases within endogenous biological contexts such as developmental studies, animal disease models, clinical samples, and other biological contexts.

## Results

### An APOBEC1-nanobody fusion protein mediates targeted RNA editing in situ

In the STAMP method, a chimeric fusion protein of APOBEC1 and an RBP-of-interest is overexpressed in cells, resulting in location-specific cytosine-to-uracil (C-to-U) RNA editing, enabling identification of authentic RBP binding sites by high-throughput sequencing from

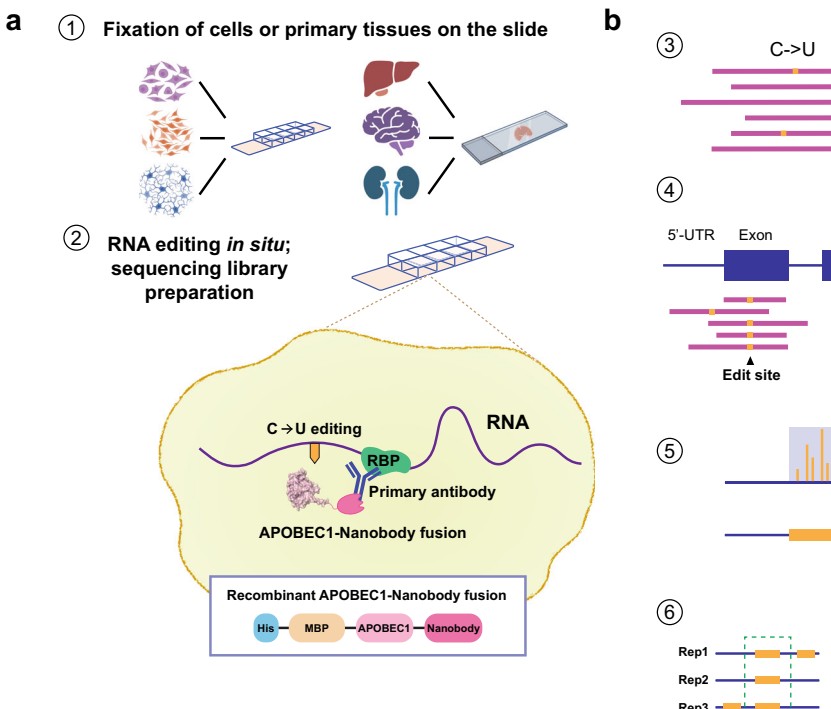

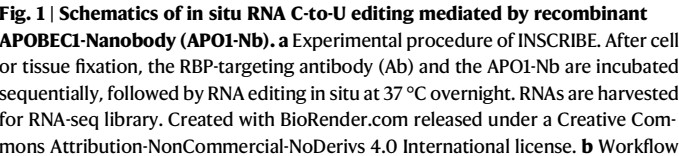

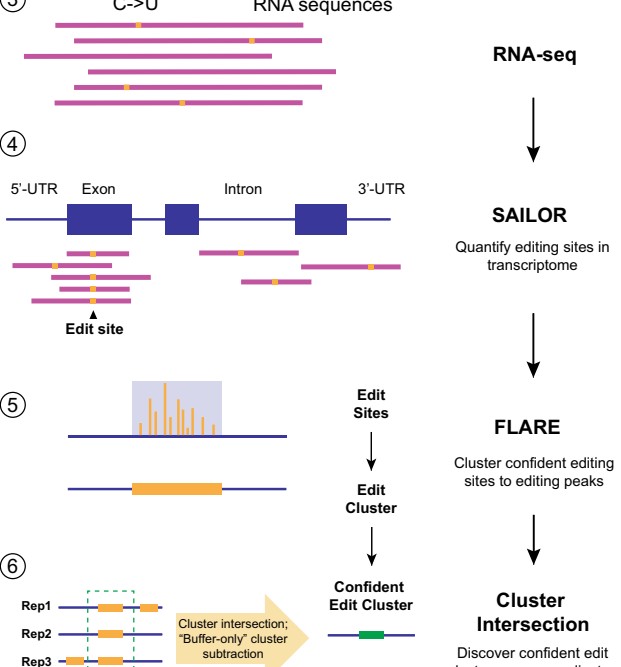

**Fig. 1 | Schematics of in situ RNA C-to-U editing mediated by recombinant APOBEC1-Nanobody (APO1-Nb). a** Experimental procedure of INSCRIBE. After cell or tissue fixation, the RBP-targeting antibody (Ab) and the APO1-Nb are incubated sequentially, followed by RNA editing in situ at 37 °C overnight. RNAs are harvested for RNA-seq library. Created with BioRender.com released under a Creative Commons Attribution-NonCommercial-NoDerivs 4.0 International license. **b** Workflow describing data analysis for INSCRIBE. The RNA sequences are aligned to the transcriptome, followed by C-to-U edit quantification by SAILOR and edit clusters identification by FLARE. Subsequently, confident edit clusters are determined by intersecting clusters from three replicates and subtracting the Buffer-only control clusters. UTR: untranslated region.

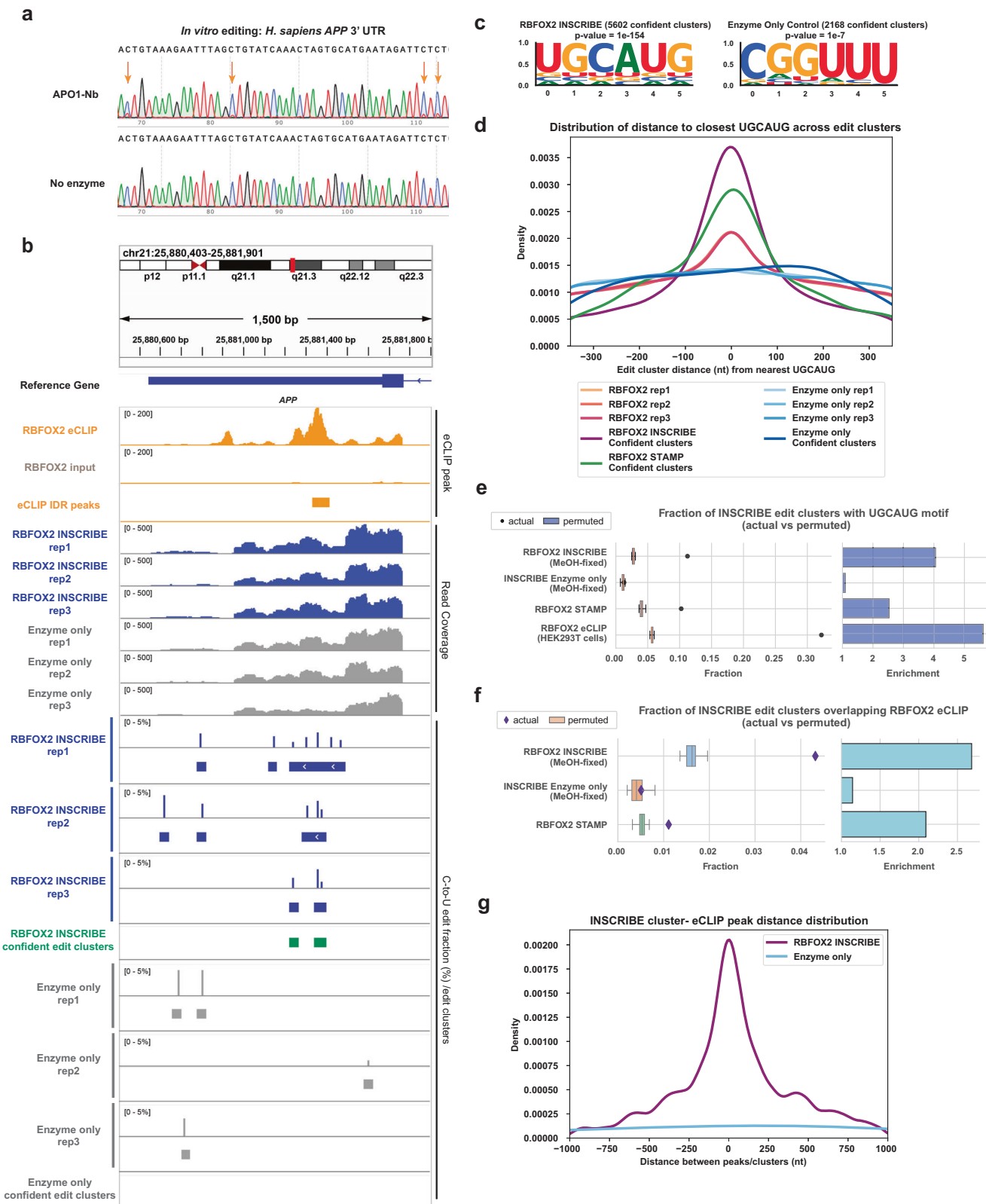

cellular RNAs[12]. We sought a method to circumvent the need to engineer individual RBP-APOBEC1 fusions in cells of interest. Since APOBEC1 is also active in vitro[14], we designed a fusion of APOBEC1 to a llama-derived single-domain antibody (nanobody) against rabbit IgG (Apo1-Nb; Fig. 1a)[15] and expressed the fusion protein in *E. coli*. A multi-step purification protocol yielded a > 95% pure and homogeneous species with a molecular weight consistent with dimeric Apo1-Nb

(Supplementary Fig. 1a, b). We verified the deaminase activity of Apo1-Nb by incubating the protein with an in vitro-transcribed 300 nt RNA derived from the 3′-UTR of the human *APP* gene (Supplementary Data 1), resulting in ~25% C-to-U editing at multiple sites as assayed by Sanger sequencing (Fig. 2a). Notably, we observed no editing in high-salt buffers (75 mM or 300 mM NaCl) or at low temperature (25 °C), permitting precise control of the editing reaction and minimization of

**Fig. 2 | INSCRIBE identifies authentic RBFOX2 binding sites through confident C-to-U edit clusters. a** Sanger sequencing result of the in vitro RNA editing assay validated the C-to-U editing enzymatic activity of the recombinant APOBEC1-nanobody (APO1-Nb). UTR: untranslated region. **b** Integrative genome viewer (IGV) tracks showing RBFOX2 eCLIP peaks (HEK293T) on the target gene *APP* expanding a 1500 bp window, compared with the read coverage, SAILOR-quantified edit fraction and FLARE-determined edit clusters of three replicates of RBFOX2-INSCRIBE, along with the enzyme-only negative controls. The confident edit clusters derived from three replicates were also shown. **c** HOMER de novo motif discovery identified the canonical RBFOX2 motif UGCAUG as the top motif in all RBFOX2-INSCRIBE replicates, using a cumulative hypergeometric distribution for *p* values. **d** Distribution of the distance between the closest UGCAUG and confident edit clusters for RBFOX2-INSCRIBE replicates (orange) and confident edit clusters (violet), enzyme-only control replicates (light blue) and confident edit clusters (dark blue), and RBFOX2-STAMP (green). A peak at 0nt distance indicated the enrichment of UGCAUG in proximity to the edit cluster center, demonstrating the

RBFOX2-binding driven editing. The RBFOX2-INSCRIBE confident edit clusters presented much stronger enrichment of adjacent UGCAUG motif than the individual replicates. **e, f** The actual fraction of UGCAUG-containing RBFOX2-INSCRIBE edit clusters/RBFOX2 eCLIP IDR peaks (dots) compared to the 20 permuted clusters (box plot, *n* = 20). Z-scores: RBFOX2-INSCRIBE, 49.93; Enzyme-only control, 0.43; RBFOX2-STAMP, 23.08; RBFOX2-eCLIP, 117.83. **f** The actual fraction of eCLIP-overlapping RBFOX2-INSCRIBE edit clusters (diamonds) compared to the permuted clusters (box plot, *n* = 20). Z-scores: RBFOX2-INSCRIBE, 18.84; Enzyme-only control, 0.39; RBFOX2-STAMP, 5.45. In (**e**–**f**), the box shows the quartiles while the whiskers extend to show the rest of the distribution; the median is represented by the center line. Enrichment is defined as the ratio of actual fraction to the mean of the permuted-derived fractionThe actual fraction was plotted using the confident edit clusters derived from 3 INSCRIBE technical replicates. **g** Distribution of the distance between the closest eCLIP peak and confident edit clusters for RBFOX2-INSCRIBE (violet) and enzyme-only control (light blue).

---

non-specific RNA editing (Supplementary Fig. 1c). Immunostaining showed that Apo1-Nb was recruited to the nucleus of methanol-fixed HEK293T cells in the presence of a rabbit-derived antibody against RBFOX2 (Supplementary Fig. 1d–f). Through immunostaining, we systematically varied the concentrations of both the anti-RBFOX2 primary antibody and Apo1-Nb to optimize the targeting specificity, as measured by the nuclear fraction of RBFOX2 localization and the co-localization of the anti-RBFOX2 antibody and Apo1-Nb. We chose a 1:200 dilution of anti-RBFOX2 Ab and a 1:1000 dilution of APO1-Nb to maximize signal intensity (Global Intensity) while maintaining specificity (Nuclear Fraction). Thus, Apo1-Nb is an active RNA base editor that can be specifically recruited to sites of interest in fixed cells by primary antibodies.

We next carried out in situ RNA editing in methanol(MeOH)-fixed HEK293T cells, since methanol fixation largely maintains RNA integrity[16]. HEK293T cells were fixed on a slide and incubated with a rabbit-derived anti-RBFOX2 antibody. After washing, APO1-Nb was immobilized to the antibody on ice in phosphate buffered saline + Tween-20 (PBS-T) buffer, where the low temperature and high salt concentration (-175 mM) prevented non-specific RNA editing. Slides were then incubated overnight at 37 °C in a buffer containing 20 mM NaCl to allow RNA editing, followed by RNA extraction and library preparation for RNA-seq. Multiple control samples were incorporated, including "Enzyme-only" which lacked the anti-RBFOX2 primary antibody, and "Buffer-only" which lacked both the anti-RBFOX2 antibody and Apo1-Nb.

We adapted the STAMP computational pipeline to identify clusters of C-to-U editing sites representing RBP-RNA interaction sites in the transcriptome from INSCRIBE samples. In brief, C-to-U editing events were identified by transcriptomic mapping and quantification through a modified SAILOR pipeline[12,17], followed by edit cluster identification through FLARE (FLagging Areas of RNA-editing Enrichment)[18] (Fig. 1b). FLARE builds on the edit site outputs of the SAILOR algorithm to identify regions that are statistically enriched for RNA editing. Based on the enzyme efficiency that we observed in the in vitro RNA editing assay, we filtered out editing sites with a SAILOR score <0.5 (*P* value > 0.5) or an editing fraction >80% (which likely represent cell line specific SNPs) before proceeding to FLARE, which retained 75%–80% SAILOR C-to-U edit sites. With -60 million reads per replicate, the pipeline typically identified around 200,000 edit clusters across the transcriptome in each replicate for RBFOX2-INSCRIBE, while roughly 120,000 edit clusters were determined in controls (Supplementary Data 2). Sites called in buffer-only controls represent C-to-U conversions introduced by RNA damage, reverse transcription errors, or PCR errors during the experimental protocol and library preparation. Nonetheless, all three replicates of RBFOX2-INSCRIBE showed clustered C-to-U editing sites at the 3′-untranslated region (3′-UTR) of the known RBFOX2 target mRNA *APP*, whereas enzyme-only controls

showed few or no editing sites in this region (Fig. 2b). The canonical RBFOX2 binding motif UGCAUG was enriched in de novo motif discovery[19] in all replicates of RBFOX2-INSCRIBE (Supplementary Fig. 1g), demonstrating the specificity of this method for discovering authentic RBP binding sites. Edit fractions across replicates in transcriptomic regions of RBFOX2-INSCRIBE edit clusters exhibited a strong correlation (Pearson R = 0.83 and 0.81, Supplementary Fig. 2a), indicating that INSCRIBE has strong reproducibility.

We further investigated the effect of APO1-Nb concentration on the specificity of INSCRIBE by measuring enrichment of the canonical RBFOX2 motif (UGCAUG) in edit clusters (Supplementary Fig. 2b). The best enrichment of proximal UGCAUG in edit clusters was observed in 1:1000 APO1-Nb dilution ratio (corresponding to a final concentration of 1 μg/mL). Higher or lower concentrations of APO1-Nb led to reduced RBP probing specificity, due to either insufficient Apo1-Nb and suboptimal editing efficiency, or oversaturated APO1-Nb and increased non-specific editing. This result agrees with our immunofluorescent staining observations (Supplementary Fig. 1d–f), and we therefore concluded that 1 μg/mL APO1-Nb is optimal for RBFOX2-INSCRIBE.

Since the number of edit clusters identified in both buffer-only and enzyme-only negative controls were nontrivial, we applied further background noise elimination strategies (Fig. 1b). First, only edit clusters that appeared in all three experimental replicates were retained. Second, to reduce experimental noise introduced by endogenous C-to-U editing in cells, edit clusters identified in buffer-only controls were removed. These two steps removed 97-98% of edit clusters in each individual replicate, and the remaining edit clusters were termed "confident edit clusters" and used for further analysis (Supplementary Data 3). FLARE identified 5,602 confident edit clusters in RBFOX2-INSCRIBE in methanol-fixed HEK293T cells, while 2168 confident edit clusters were determined in enzyme-only controls. The canonical RBFOX2 motif was enriched in RBFOX2-INSCRIBE confident edit clusters (*p* = 1e−154), while it was not enriched in both enzyme-only and buffer-only controls (Fig. 2c). Confident edit clusters after these noise-reduction strategies showed higher enrichment of UGCAUG compared to edit clusters identified in individual replicates, demonstrating enhanced specificity of RBFOX2 target RNA identification (Fig. 2d).

Further examination of the confident edit clusters showcased the accuracy of RBP binding sites discovered by INSCRIBE. We found that around 10% of confident RBFOX2-INSCRIBE edit clusters contained the UGCAUG motif, 4-fold enriched over the randomly shuffled (permuted) size-matched clusters within the same regions (Fig. 2e), which was comparable to RBFOX2-STAMP[18]. In contrast, confident edit clusters in enzyme-only controls showed no enrichment of the UGCAUG motif over permuted clusters, indicating the randomness of editing in the absence of the anti-RBFOX2 primary antibody. Empirically defined RBFOX2 sites were defined as reproducible peaks from irreproducible

discovery rate (IDR) analysis of HEK293T RBFOX2 eCLIP data and were compared to RBFOX2-INSCRIBE. We observed that the fraction of RBFOX2-INSCRIBE confident edit clusters overlapping with RBFOX2-eCLIP IDR peaks was enriched (>2.5 fold) compared to permuted clusters (Fig. 2f) and that INSCRIBE clusters were enriched near eCLIP IDR peaks (Fig. 2g), indicating that INSCRIBE identifies true RBFOX2 binding sites. Examination of individual edit clusters of INSCRIBE revealed their proximity and concurrence with eCLIP peaks (Supplementary Fig. 2c). In summary, we established an optimal working protocol for INSCRIBE by APO1-Nb RNA editing in MeOH-fixed HEK293T cells, which enabled the successful identification of RBFOX2 binding sites in situ.

## Comparison of INSCRIBE and eCLIP reveals differences in resolution

Next, we focused on how RBFOX2-INSCRIBE confident clusters compares to the binding sites identified by conventional gold-standard method eCLIP. Despite the low overlap (Fig. 2f) of INSCRIBE edit clusters with eCLIP binding sites, we observed, expectedly, that RBFOX2-INSCRIBE clusters that overlap with RBFOX2-eCLIP peaks had enriched UGCAUG motifs (Supplementary Fig. 3a). Interestingly, INSCRIBE clusters that did not have an overlapping eCLIP peak also exhibited statistically significant ($p$ value < $10^{-111}$) UGCAUG motifs (Supplementary Fig. 3b). When we studied the types of genic regions found to contain overlapping or non-overlapping clusters, we observed distinct preferences. (Supplementary Fig. 3c). Compared to eCLIP, INSCRIBE discovered a lower fraction of edit clusters within introns and a higher fraction within 3′UTRs, likely due to the fundamental difference in selection and read coverage for the different regions within pre-mRNAs. Specifically, introns were not as deeply covered by reads and without a positive antibody selection as in eCLIP, intronic regions were less represented in the INSCRIBE data. Nevertheless, despite the lower-than-anticipated overlap between INSCRIBE edits and eCLIP binding sites due to differences in regional representation, our results indicate that authentic binding sites are being recovered by INSCRIBE.

Another difference between eCLIP and INSCRIBE is the resolution of identified binding sites, as the APOBEC1 enzyme recognizes the RBP through an antibody-nanobody linkage. We expected that INSCRIBE edit clusters might reside at a distance from the actual nucleotides that interacted with the RBP. As RBFOX2 interacts with the UGCAUG motif, we plotted the cumulative distribution of INSCRIBE clusters or enzyme-only clusters with increased distance to UGCAUG (Supplementary Fig. 3d). The slope of the enzyme-only control curve reflected the variation of APOBEC1 editing and the natural occurrence of UGCAUG in the transcriptome. We observed RBFOX2-INSCRIBE converges with enzyme-only at ~200nt from UGCAUG (Supplementary Fig. 3e), which indicates that the INSCRIBE edit radius is at most 200nt from the true binding site. This likely also contributed to a lower overlap with eCLIP binding sites than expected if the edit cluster was computationally defined outside of the ~50-100 nt eCLIP-defined windows. Therefore, we observed a slightly lower enrichment of UGCAUG motifs within RBFOX2-INSCRIBE clusters compared to eCLIP-defined sites. Nevertheless, in summary, expanding the RBFOX2 motif to GCAYG (Y = C or U) and including secondary motifs GCUUG, GAAUG, GUUUG, GUAUG, GUGUG and GCCUG as defined in previous literature[20], around 54% of RBFOX2-INSCRIBE confident edit clusters have either eCLIP or motif-based evidence for RBFOX2 interactions.

## INSCRIBE enables RNA isoform distinctions in RBP binding sites discovery

To examine whether INSCRIBE is compatible with long-read sequencing, which enables the detection of RBP binding at mRNA isoform resolution, we generated and sequenced libraries from RBFOX2-INSCRIBE and the enzyme-only control using the PacBio sequencing platform. We chose the PacBio system as it currently has a lower base-calling error rate than the Oxford Nanopore Technologies platform[12,21]. After removing SNPs (Methods), de novo motif discovery within edited regions confirmed that both replicates of long-read RBFOX2-INSCRIBE exhibited statistically significant enriched ($p$ value = 1e−98 and 1e−102 for replicate1 and replicate2) UGCAUG motifs (Fig. 3a). Additionally, the edits detected at the mRNA isoform level showed a high degree of correlation between replicates as measured using editsC (ratio of number of edited Cs relative to the number of Cs across exons and UTRs of isoforms), with a Pearson correlation R = 0.95 (Supplementary Fig. 4a), consistent with the high reproducibility of INSCRIBE experiments.

RBFOX2-INSCRIBE displayed general higher C-to-U editing rates than enzyme-only control (Fig. 3b). We conducted principal component analysis (PCA) using isoform-level editsC and reads per kilobase of transcript per million reads mapped (RPKM) independently to ensure that signal identified in RBFOX2-INSCRIBE was not due to differences in isoform expression. PCA showed a clear separation between RBFOX2-INSCRIBE and enzyme-only control with editsC, but not RPKM, confirming that signal differences were due to editing (Supplementary Fig. 4b). To evaluate isoform-specific RBP binding, we first filtered out RNA isoforms with read coverage <20 across replicates and edit fraction <0.02 to improve RBFOX2-INSCRIBE signal detection over background, as represented by the enzyme-only control (Supplementary Fig. 4c). We constructed a linear model using the average isoform-level editsC calculated from RBFOX2-INSCRIBE and enzyme-only samples and selected high confidence isoforms in RBFOX2-INSCRIBE samples (>1.5 std.) for subsequent analysis (Orange in Supplementary Fig. 4d, $n$ = 204).

Examples of differential isoform editing signatures are illustrated by *DCUN1D4* and *DEPDC1* genes (Fig. 3c, d and Supplementary Fig. 4e). These differential isoform edits agree with both RBFOX2 eCLIP and short-read RBFOX2-INSCRIBE edit clusters. We conclude that INSCRIBE enables isoform-sensitive identification of RBP-RNA interactions when coupled with long-read sequencing.

## INSCRIBE identifies RBP binding sites with low sequencing depth and ultra-low input RNA

As the accuracy of C-to-U edit cluster calls depends on the sequencing read depth of the RNA in question, it is potentially challenging to identify RBP-RNA interactions on lowly expressed transcripts or in intronic regions with low read coverage. To evaluate the sequencing depth required by INSCRIBE for reliable edit cluster identification, we sequenced short-read libraries from RBFOX2-INSCRIBE and controls from methanol-fixed HEK293T cells to roughly 100 million (100 M) reads per replicate, then randomly down-sampled each dataset to 80, 60, 40, and 20 million reads. The number of C-to-U edit clusters determined by FLARE was proportionally associated with sequencing depth in each sample (Supplementary Fig. 5a, Supplementary Data 4). A greater number of confident edit clusters were identified with higher sequencing depth, but the canonical RBFOX2 motif was identified in all samples regardless of sequencing depth through de novo motif analysis (Supplementary Fig. 5b). Enrichment of the UGCAUG motif was generally higher in deeper-sequenced data, although the difference between 100 M reads and 80 M reads was negligible (Fig. 4a and Supplementary Fig. 6a). As expected, increased sequencing depth resulted in an increased count and proportion of identified intronic edit clusters (Supplementary Fig. 5d). With 100 M reads, 61.5% (5938 counts) of the confident edit clusters fell into intronic regions, compared to 21.5% (285 counts) with 20 M reads (Supplementary Data 4). As a splicing regulator, RBFOX2 exhibits positional dependencies in its RNA binding profile, binding upstream to included exons and downstream to excluded exons[22] revealed by eCLIP (Supplementary Fig. 5e). We demonstrated INSCRIBE could discover the splicing regulatory binding patterns of an RBP only with deeper sequencing depths (100 M

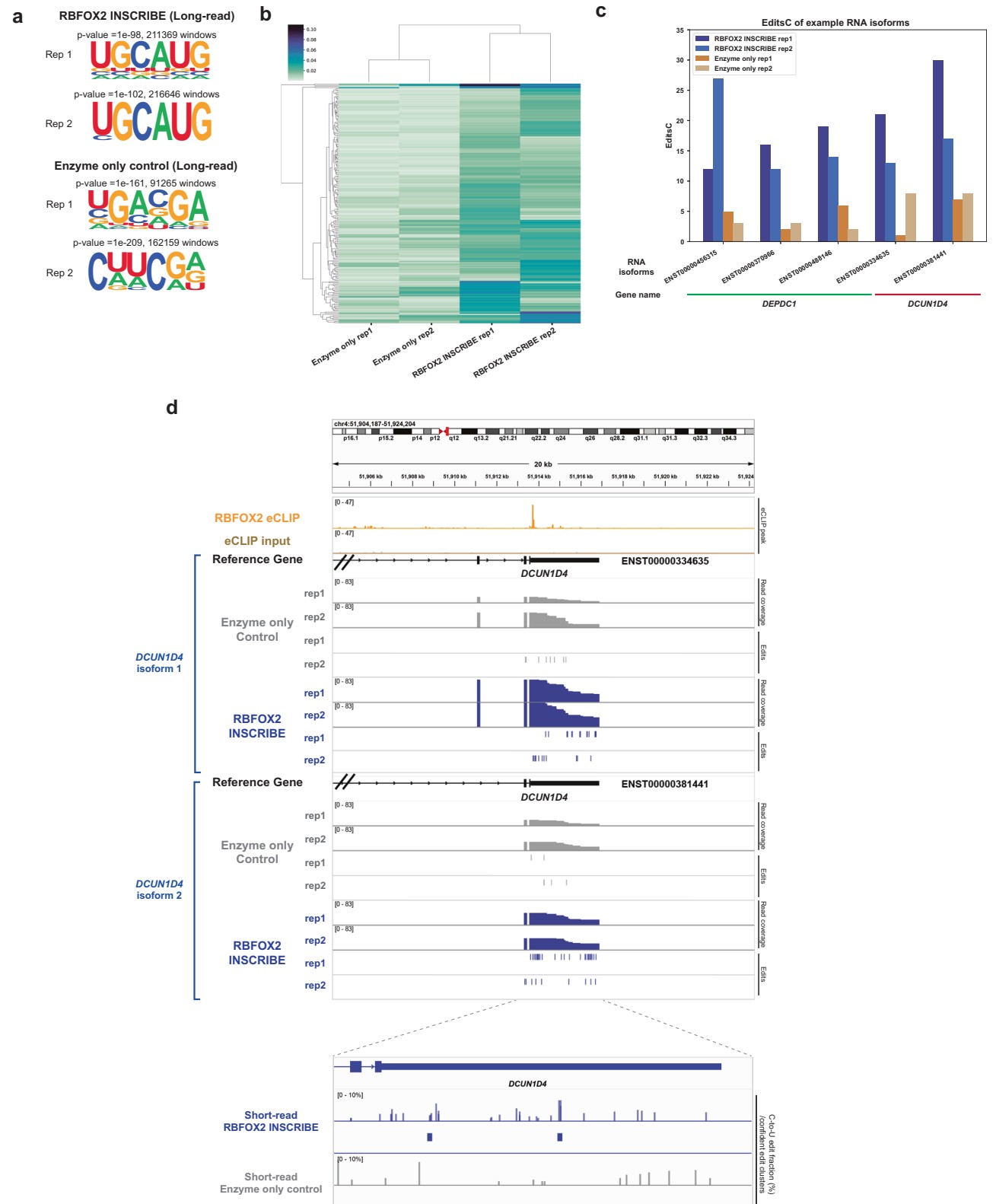

**Fig. 3 | Long-read sequenced INSCRIBE enables mRNA isoform distinctions.**
**a** HOMER de novo motif discovery of PacBio sequenced INSCRIBE replicates identified the canonical RBFOX2 motif (UGCAUG), using a cumulative hypergeo-metric distribution for *p* values. **b** Heatmap of editsC (ratio of number of edited Cs relative to the number of Cs across exons and UTRs of isoforms) of individual RNA isoforms from both replicates of RBFOX2-INSCRIBE and enzyme-only controls. **c** EditsC of example RNA isoforms in gene *DEPDC1* and *DCUN1D4* from both replicates of RBFOX2-INSCRIBE and enzyme-only control. **d** Integrative genome viewer (IGV) tracks showing PacBio sequenced RBFOX2-INSCRIBE vs enzyme-only control edits on two isoforms of *DCUN1D4* in a 20 kb window, alongside with INSCRIBE read coverage, eCLIP peaks (HEK293T). The short-read sequenced RBFOX2-INSCRIBE vs enzyme-only control edit fractions and the confident edit clusters are shown at the bottom panel for comparison.

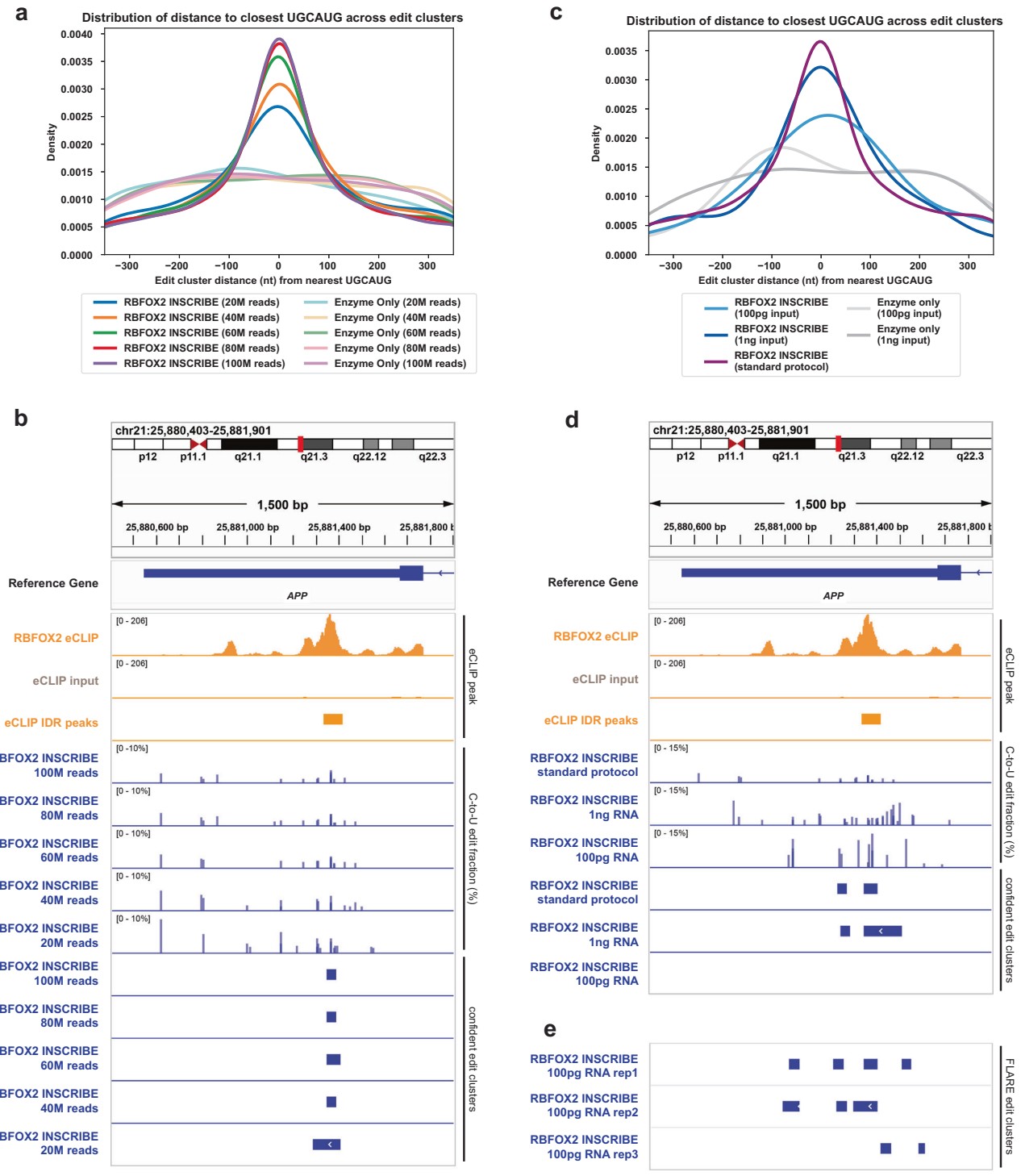

**Fig. 4 | The performance of INSCRIBE at various sequencing depths and ultra-low RNA input. a** The enrichment of UGCAUG motif at confident edit clusters center with different sequencing depths of RBFOX2-INSCRIBE and enzyme-only control. Increasing sequencing depth from 20 to 40, 60, 80, 100 million (M) reads enhanced the enrichment of UGCAUG, while the improvement from 80 M to 100 M was marginal. **b** Integrative genome viewer (IGV) tracks of representative RBFOX2 target gene (*APP*) examining the effect of sequencing depth on SAILOR-quantified C-to-U conversion pattern and FLARE confident edit clusters. The tracks include eCLIP peaks (orange), 3-replicate overlay of SAILOR-quantified edit fraction (blue, upper tracks) and FLARE confident edit clusters (blue, lower tracks) of RBFOX2-INSCRIBE. **c** The enrichment of UGCAUG motif at confident edit clusters center with

libraries of different input total RNA of RBFOX2-INSCRIBE and enzyme-only controls. Both 1 ng (dark blue) and 100 pg (light blue) input RNA showed evident enrichment of UGCAUG motif as compared to standard input (>1000 ng RNA) (purple) in RBFOX2-INSCRIBE, while lower amount of input material decreased the enrichment. **d** IGV tracks of representative RBFOX2 target gene (*APP*) examining the effect of input starting total RNA on SAILOR-quantified C-to-U conversion pattern and FLARE confident edit clusters. The tracks include eCLIP peaks (orange), 3-replicate overlay of SAILOR-quantified edit fraction (blue, upper tracks) and FLARE confident edit clusters (blue, lower tracks) of RBFOX2-INSCRIBE. **E** Continued IGV tracks showing the FLARE-identified edit clusters in individual replicates for the 100 pg input RNA libraries.

reads), showing a significant enrichment in downstream intron to excluded exons and partly reproducing the eCLIP results. The less sequenced INSCRIBE did not capture such patterns, which is likely due to the inadequate intronic binding sites discovered with 60 M reads (Supplementary Fig. 5e). At the level of individual sites, deeper sequencing depth effectively reduced some inflated C-to-U edit fractions and thereby reduced the noise arising from low read coverage (Fig. 4b). Nonetheless, both the overall SAILOR-quantified edit site patterns and the FLARE confident edit clusters remained similar across all samples from 20 M to 100 M reads (Fig. 4b), as opposed to the enzyme-only controls (Supplementary Fig. 6c). Irrespective of sequencing depth, INSCRIBE showed a similar level of concordance with eCLIP IDR peaks (Supplementary Fig. 6b). In conclusion, we observed that INSCRIBE demonstrates effective identification of RBP binding sites with sequencing depth as low as 20 M reads. For targets such as intronic sites or low-level transcripts, deeper sequencing depths is helpful, but it is unnecessary to exceed 100 M reads per sample.

Next, we sought to determine the detection limit of INSCRIBE, as the input material requirement has been the major hurdle of profiling RBP-RNA interactions in precious tissues such as clinical samples or stem cell-derived models. In the standard RBFOX2-INSCRIBE experiment in HEK293T cells, more than 1000 ng of total RNA is typically extracted from ~150,000 cells. Aware of the cost per experiment for valuable samples, we scaled down the amount of total RNA as input over 1000-fold, to either 1 ng or 100 pg, and prepared RNA-seq libraries. Notably, with ~50 M sequencing depth, both 1 ng and 100 pg input showed promising enrichment of the RBFOX2 canonical motif UGCAUG (Fig. 4c and Supplementary Fig. 5f). The ultra-low input RBFOX2-INSCRIBE also showed robust performance in recovering the UGCAUG motif from only 776 edit clusters from 1 ng total RNA and 214 clusters from 100 pg total RNA (Supplementary Fig. 5c). Like the INSCRIBE library with lower sequencing depth, ultra-low input INSCRIBE unavoidably presented inflated C-to-U edit fractions when mapped onto the transcriptome (Fig. 4d). Nonetheless, the SAILOR-quantified edits still cluster around eCLIP peaks in contrast to the enzyme-only controls (Supplementary Fig. 6d). For 100 pg input RNA, FLARE successfully discovered this binding site in all three individual replicates (Fig. 4e). We recognized that some RNA targets might be dropped out with 100 pg of input RNA, such as the known binding site within the 3′UTR in the *APP* gene, which is a common caveat of low-input or single cell sequencing technologies, but INSCRIBE can still identify accurate RNA targets of RBFOX2. Both ultra-low input libraries showed significant enrichment of overlap with eCLIP peaks in INSCRIBE-discovered edit clusters (Supplementary Fig. 5g). Therefore, we demonstrate high sensitivity of INSCRIBE and showcase the transcriptomic discovery of endogenous RBP-RNA interactions from as little as 100 pg of input RNA, which is equivalent to five cells as the starting material.

### INSCRIBE identifies authentic TDP-43 binding sites through confident C-to-U edit clusters

To demonstrate the versatility of the APO1-Nb design and INSCRIBE's performance in profiling other RBPs, we optimized the dilution factor for a rabbit-derived anti-TDP-43 antibody and successfully profiled TDP-43, a nuclear RNA/DNA-binding protein implicated in many neurodegenerative diseases[23]. FLARE identified 5,751 confident edit clusters and de novo motif discovery on confident edit clusters of TDP-43-INSCRIBE showed enrichment of the canonical TDP-43 binding motif UGUGUG ($p = 1e-20$) (Fig. 5a). The fraction of confident TDP-43-INSCRIBE edit clusters exhibiting a UGUGUG motif is notably higher than in shuffled size-matched peaks (Fig. 5b). The motif was also enriched in proximity of TDP-43-INSCRIBE edit clusters relative to enzyme-only controls (Fig. 5c). Moreover, we noted a more than 2.5-fold enrichment in the proportion of TDP-43-INSCRIBE peaks

overlapping with TDP-43 eCLIP, compared to enzyme-only controls (Fig. 5d). Known TDP-43 binding sites on the *TARDBP*, *NUCKS1*, *XPO1* and *FUS* genes were clearly captured by INSCRIBE C-to-U edit clusters and showed agreement with eCLIP peaks (Fig. 5e). In conclusion, INSCRIBE demonstrates its versatility in robust RNA target identification of RBPs with a universally adaptable protocol.

### INSCRIBE is compatible with multiple cell fixation methods

As formalin or formaldehyde fixation is more prevalent than methanol fixation in clinical sample collection workflows[24], we next aimed to implement INSCRIBE after formaldehyde fixation, which is known to damage RNAs and induce RNA modifications including nonspecific cytosine deamination[25]. We applied a mild de-crosslinking protocol that effectively preserved RNA integrity from paraformaldehyde (PFA) fixation (Supplementary Fig. 7a). We observed comparable numbers of confident edit clusters in PFA-fixed HEK293T cells as in methanol-fixed cells for both RBFOX2 (2,970 clusters) and TDP-43 (2,540 clusters). De novo motif discovery identified both the canonical RBFOX2 motif ($p = 1e-78$) and TDP-43 motif ($p = 1e-5$; 2nd top motif). These motifs were enriched in regions adjacent to confident edit cluster centers comparable to methanol-fixed INSCRIBE (Fig. 6a, b). We also observed a similar enrichment of the canonical motif and of the eCLIP IDR peak overlap in confident edit clusters in PFA-fixed samples as in MeOH-fixed samples (Fig. 6c, d). In PFA-fixed INSCRIBE, the C-to-U editing and the confident edit clusters at representative target genes also uncovered their striking proximity and alignment with eCLIP peaks (Fig. 6e, Supplementary Fig. 7b). In summary, INSCRIBE is compatible with PFA-fixed cells, enhancing its utility in analysis of clinical samples and extending its adaptability to a broader range of experimental workflows.

### INSCRIBE identifies RBP-RNA interaction sites in fixed tissue samples

Finally, we sought to determine whether INSCRIBE could capture authentic RBP-RNA interactions in primary tissue samples, in which elements like the extracellular matrix could inhibit the binding of a primary antibody to its target RBP, the recognition of the primary antibody by Apo1-Nb, or the editing reaction. To test INSCRIBE's compatibility with primary tissues, we conducted RBFOX2-INSCRIBE in both methanol-fixed and PFA-fixed 20 μm-thick mouse brain slices (Fig. 7a). Remarkably, INSCRIBE displayed exceptional performance when applied to mouse brain tissue. The canonical RBFOX2 motif UGCAUG was found in both the methanol-fixed and PFA-fixed INSCRIBE confident edit clusters through de novo motif discovery (Fig. 7b). The RBFOX2 motif was also enriched across all confident edit clusters compared to enzyme-only controls, with around 17% of edit clusters bearing a UGCAUG motif in the methanol-fixed samples and around 10% in the PFA-fixed samples (Fig. 7c, d). Furthermore, approximately 13% or 8% of confident edit clusters overlapped RBFOX2 eCLIP peaks in methanol-fixed or PFA-fixed mouse whole brain tissue, with a >3-fold enrichment compared to permuted clusters in both samples (Fig. 7e). Additionally, upon closer examination at individual RBFOX2 target genes, a notable correlation was observed between INSCRIBE confident edit clusters and eCLIP IDR peaks, indicating a strong alignment between the two (Fig. 7f and Supplementary Fig. 8a). In conclusion, INSCRIBE can capture RBP-RNA interactions in primary tissues, opening the possibility of its application to diverse precious clinical samples.

### Discussion

Here we describe INSCRIBE, a convenient approach to transcriptomic RBP-RNA interaction profiling with low-input material requirement and a versatile workflow. By harnessing a recombinant APOBEC1-nanobody fusion protein, in situ RNA labeling through APOBEC1-mediated cytosine deamination is directed towards RNAs bound by

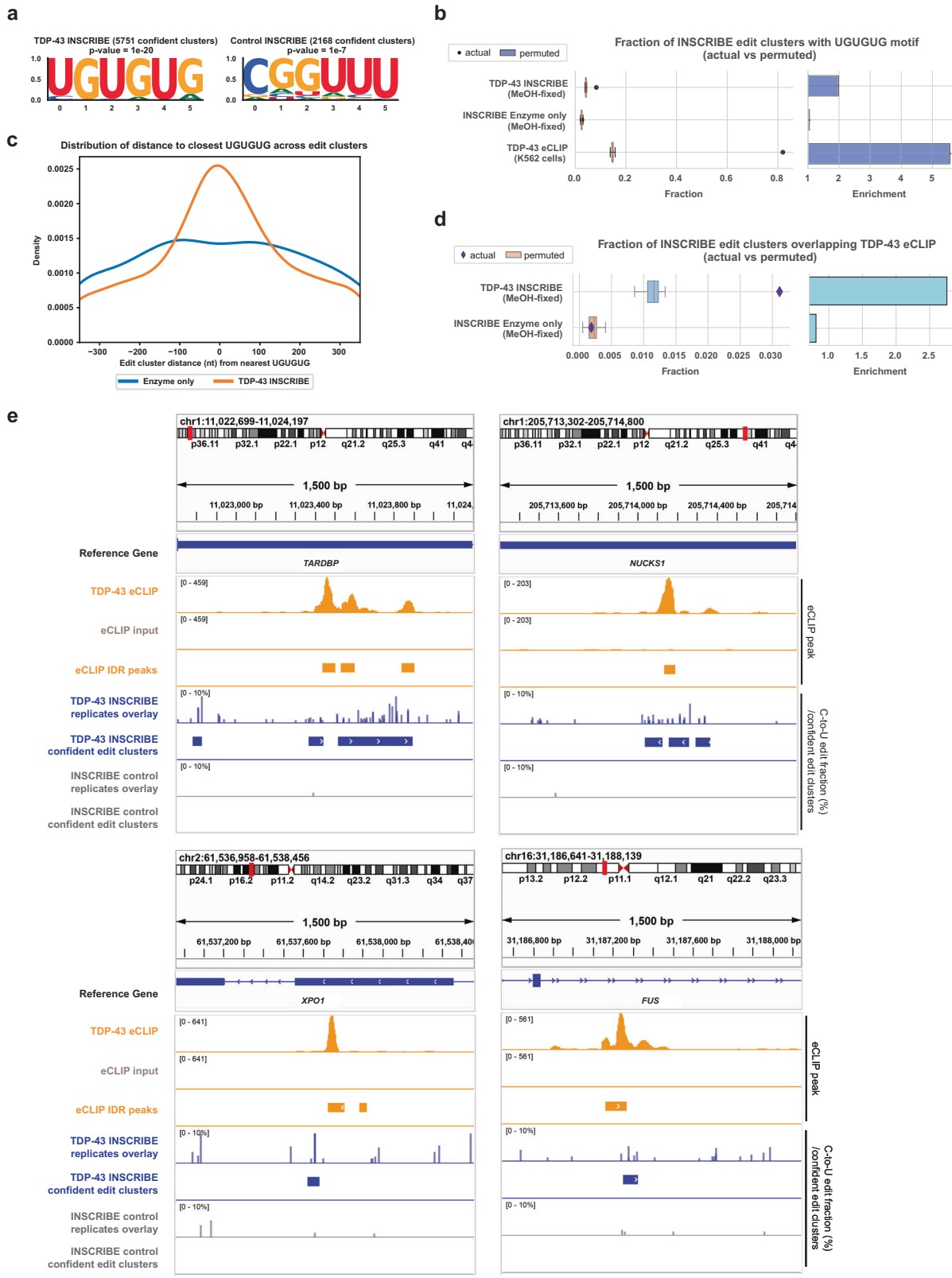

the RBP of interest. INSCRIBE is optimized for fixed cells or primary tissues, without any prior requirement for plasmid construction, transfection, or exogenous overexpression in cells or tissues. This strategy allows for the mapping of endogenous binding sites for RBFOX2 and TDP-43 transcriptome-wide in both methanol- and PFA-fixed HEK293T cells, as well as intact primary tissue samples—mouse brain tissue, with high accuracy and specificity.

Unlike conventional immunoprecipitation-based methods, we show that INSCRIBE seamlessly integrates with long-read sequencing, enabling discrimination of isoform-specific RBP-RNA interactions. Since a single APOBEC1-nanobody fusion recognizes the IgG element of all primary antibodies derived from a single animal system (rabbit in the current implementation), the method is compatible with many commercially available anti-RBP primary antibodies. The method's

**Fig. 5 | INSCRIBE identifies authentic TDP-43 binding sites through confident C-to-U edit clusters. a** HOMER de novo motif discovery of TDP-43 INSCRIBE replicates identified the canonical TDP-43 motif (UGUGUG), using a cumulative hypergeometric distribution for *p* values. **b** The actual fraction of UGUGUG-containing TDP-43-INSCRIBE edit clusters/TDP-43 eCLIP IDR peaks (dots) compared with that of the 20 permuted clusters (box plot, *n* = 20). The actual fraction was plotted using the confident edit clusters derived from 3 INSCRIBE technical replicates. The box shows the quartiles while the whiskers extend to show the rest of the distribution; the median is represented by the center line. Z-scores: TDP-43-INSCRIBE, 22.22; Enzyme-only control, 0.33; K562 TDP-43 eCLIP, 146.11. **c** The distribution of the nearest UGUGUG distance to clusters of confident edits reveals a peak at 0nt for TDP-43-INSCRIBE (orange), indicating an abundance of UGUGUG sequences in close proximity to the center of the edit cluster, which demonstrates TDP43-binding driven C-to-U editing. **d** The actual fraction of eCLIP-overlapping TDP-43-INSCRIBE edit clusters (diamonds) compared to that of the 20 permuted clusters (box plot, *n* = 20). The actual fraction was plotted using the confident edit clusters derived from 3 INSCRIBE technical replicates. The box shows the quartiles while the whiskers extend to show the rest of the distribution; the median is represented by the center line. Z-scores: TDP-43-INSCRIBE, 15.61; Enzyme-only control, −0.46. **e** Integrative genome viewer (IGV) tracks of example RBFOX2 target genes (*TARDBP*, *NUCKS1*, *XPO1*, *FUS*) with 1500 bp windows showing RBFOX2 eCLIP peaks (orange), RBFOX2-INSCRIBE (blue) and the enzyme-only control (gray). The 3-replicate overlay of SAILOR-quantified edit fraction of INSCRIBE is shown along with the FLARE-determined confident edit clusters.

versatility and user-friendliness render it conveniently applicable to complex experimental systems, such as disease-related animal models, organoids, and clinical samples. It readily bridges the critical gap in RBP-RNA profiling in RNA-related diseases under authentic endogenous biological environments, facilitating insights into a wide spectrum of biological inquiries and disease mechanisms.

Remarkably, the high sensitivity of INSCRIBE enables the transcriptome-wide capture of RBP-RNA interaction with low sequencing depth (as low as 20 million reads) or ultra-low input material (as low as 100 pg total RNA, equivalent to ~5 mammalian cells). With a typical INSCRIBE workflow, one could therefore investigate the spatial heterogeneity of RBP-RNA interactions in a tissue cost-effectively. For example, INSCRIBE experiments with tissue slices of different coordinates across the brain for RBM5, an alternative splicing regulator that binds RNA differentially in Huntington's disease mouse brain[26], could potentially reveal more detailed roles RBM5 plays in distinct brain regions, structures and cortical layers in Huntington's disease. Spatial profiling of RBP-RNA interactions with INSCRIBE would enable the scrutinization of the RNA target changes occurring in tissue microenvironments during disease development and therapeutic response. Currently, we are keen on advancing the technology for single-cell applications and spatial transcriptomics, empowering simultaneous RBP-RNA interaction and RNA localization analysis in primary tissues.

One limitation of the current INSCRIBE protocol is its reliance on highly specific anti-RBP antibodies, which are dependent on animals to produce. We anticipate that the advancement of AI(artificial intelligence)-designed nanobodies could alleviate this challenge[27]. In the meantime, a broad range of disease-relevant RBPs, including TDP-43, can already be effectively targeted with commercially available high-quality antibodies. Although current INSCRIBE utilizes rabbit-derived primary antibodies, the adaptable nanobody segment can be tailored to target antibodies from various animals, broadening its scope.

Additional challenges faced by INSCRIBE are the inherent sequence and structural preferences in RNA-modifying enzymes and background noise. The APOBEC1 enzyme more readily edits less structured RNA sequences and appears to prefer specific local sequence contexts (5′-A/U of the edited C)[28,29]. Future endeavors will focus on engineering alternative RNA-modifying enzymes with reduced substrate context preferences. We reason that a significant contributor to the background noise arises from enzymes that are not immobilized to the antibody during the RNA editing reaction. To mitigate background noise from unbound enzymes, it is essential to carefully choose antibodies with superior performance in immunofluorescence contexts, and to optimize the dilution factor for each antibody. Based on the relatively high edit cluster counts in the enzyme-only controls, we conclude that meticulous optimization of both the anti-RBP-of-interest Ab and APO1-Nb concentrations will be required to minimize background noise. This noise could be particularly problematic for intronic regions or low-level transcripts, where the sequencing read coverage is generally low. Increasing sequencing depth could be helpful if the RBP-of-interest targets intronic or low-level transcripts. We also acknowledge the challenge associated with achieving an optimal signal-to-noise ratio when profiling low-abundance RBPs. In such scenarios, it will be crucial to maintain low concentrations of both the primary antibody and APO1-Nb to minimize nonspecific editing. Future advancements in this technology will prioritize enhancing the binding specificity and affinity of the antibody-IgG-recognizing nanobody.

## Methods

### Ethical statement
The mouse work in this study was carried out in strict accordance with the recommendations in the Guide for the Care and Use of Laboratory Animals of the National Institutes of Health and approved by the Animal Care and Use Committee at University of California San Diego.

### Plasmid construction and recombinant protein purification
The amino acid sequence of rat APOBEC1 (NP_037039.1) was codon-optimized for expression in *E. coli* strain and cloned into a modified UC Berkeley Macrolab vector 2CT (Addgene #29706) to generate an N-terminal fusion to a His$_6$-MBP (Maltose-Binding Peptide) tag. The sequence of an engineered anti-rabbit-IgG nanobody (TP897, Addgene plasmid # 104163)[15] was cloned at the C-terminal of the APOBEC1 with an 18-aa linker (GSGTSGAGSATAGSGAGG) in-between. Plasmid pTP1183 was a gift from Dirk Görlich (Addgene plasmid # 104163; RRID: Addgene_104163). The His-MBP-APOBEC1-Nb (APO1-Nb) protein was expressed in *E. coli* strain Rosetta2 pLysS (EMD Millipore) by growing cultures in 2xYT media to mid-log phase at 37 °C, followed by induction with 0.25 mM IPTG at 18 °C for 16 h supplemented with 100 μM ZnCl$_2$. For protein purification, cells were harvested by centrifugation, suspended in resuspension buffer (25 mM Tris-HCl pH 8.5, 1 M NaCl, 20 mM imidazole, 2 mM β-Mercaptoethanol and 10% glycerol) with 0.1 mg/mL RNase A and lysed by sonication. Lysates were clarified by centrifugation (35,267x g 30 mins), then the supernatant was loaded onto a Ni$^{2+}$ affinity column (HisTrap HP, Cytiva) pre-equilibrated with the resuspension buffer. The column was washed with a buffer containing 20 mM imidazole and 500 mM NaCl, and eluted with a buffer containing 250 mM imidazole and 250 mM NaCl. The elution was buffer-exchanged into the low salt buffer (20 mM Tris pH8.5, 50 mM NaCl, 1 mM DTT and 10% glycerol) and loaded onto an anion-exchange column (HiTrap Q HP, Cytiva) following washing with a gradient of 50 mM-1 M NaCl. The eluted fractions containing the protein (evaluated with SDS-PAGE) were collected and loaded onto a size exclusion column (Superdex 200 Increase 10/300 GL, Cytiva) equilibrated with the size exclusion buffer (20 mM Tris pH7.5, 300 mM NaCl, 1 mM DTT and 10% glycerol). The peak containing the dimerized His-MBP-APOBEC1-Nb protein was collected concentrated by ultrafiltration, aliquoted and frozen at −80 °C for future use.

### In vitro RNA editing assay
A 300nt RNA from the 3′-UTR of the human *APP* gene was in vitro-transcribed (MEGAshortscript T7 Transcription Kit, Invitrogen) and purified (Zymo RNA Clean & Concentrator). For a typical RNA editing assay, 0.1 μM RNA was incubated with 1.2 μM APO1-Nb (or other Enzyme-Nb) at 37 °C for 16 h in RNA editing buffer (20 mM Tris-HCl at

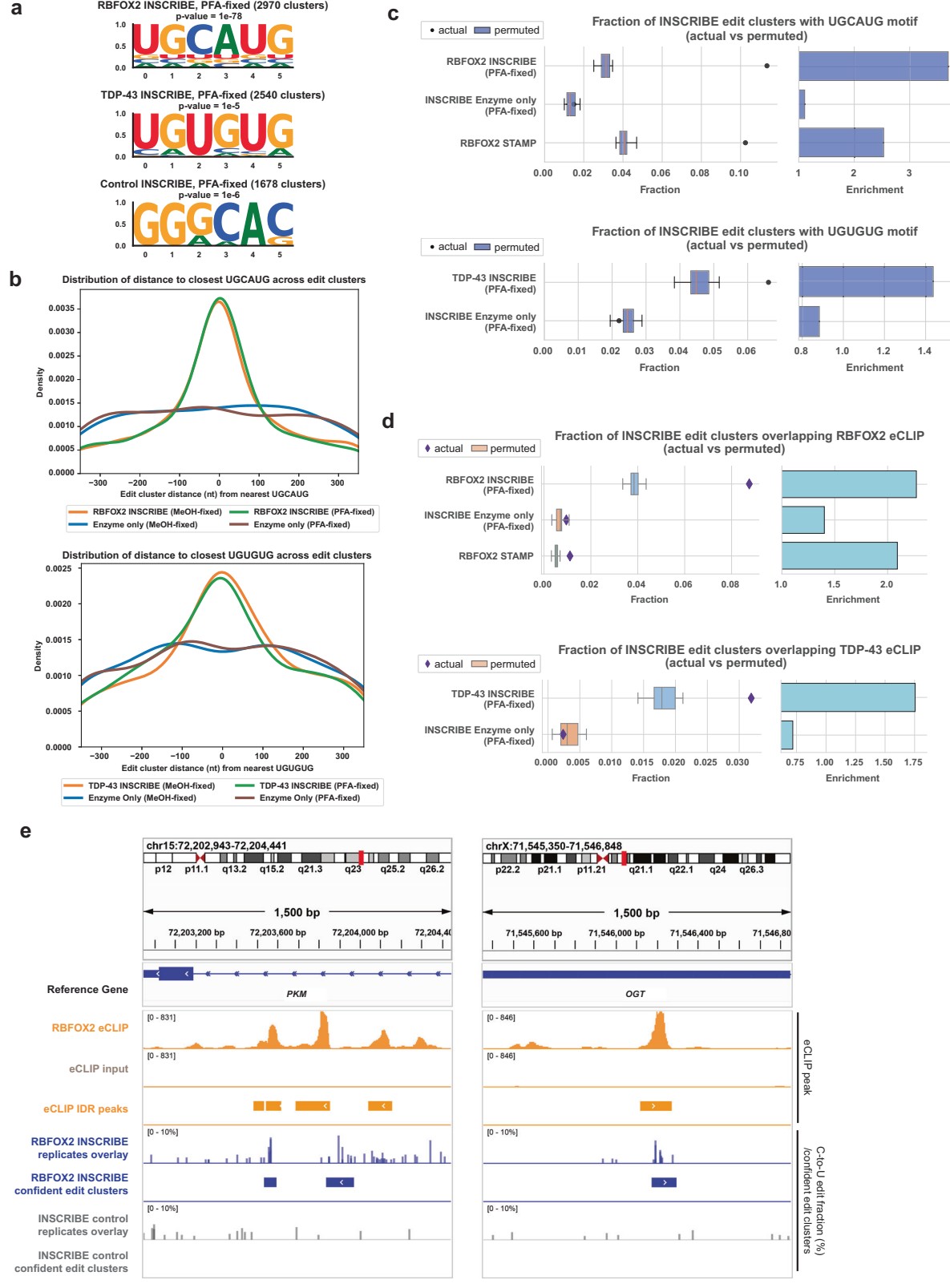

desired pH, 5 mM DTT, 2U/μL RNaseOUT, desired final concentration of NaCl). The reaction was quenched at 95 °C for 10 mins. The edited RNAs were reverse-transcribed to cDNA (High-Capacity cDNA Reverse Transcription Kit, Thermo) followed by 20-cycles of PCR amplification and Sanger sequencing (Azenta). Template and primer sequences were provided in Supplementary Data 1.

## Cell culture
HEK293T cells (Takara Bio, 632180) were cultured in DMEM medium (high glucose, Gibco) with 10% (v/v) heat inactivated fetal bovine serum FBS (Gibco) and 5 U/mL Penicillin-Streptomycin (Gibco) at 37 °C, 5% $CO_2$. All cell lines were maintained at 37 °C and 5% $CO_2$ and routinely tested for the presence of mycoplasma.

**Fig. 6 | INSCRIBE exhibited comparable performance in PFA-fixed cells.**
**a** HOMER de novo motif analysis of confident edit clusters revealed the canonical RBFOX2 motif UGCAUG (top motif) or TDP-43 motif UGUGUG (2nd top motif), for RBFOX2-INSCRIBE or TDP-43-INSCRIBE in paraformaldehyde (PFA)-fixed HEK293T cells respectively, using a cumulative hypergeometric distribution for *p* values. **b** A density plot of the distance between edit clusters to closest UGCAUG (for RBFOX2-INSCRIBE, upper panel) or UGUGUG (for TDP-43-INSCRIBE, lower panel). INSCRIBE in MeOH-fixed and PFA-fixed cells presented no obvious distinction in canonical motif enrichment at the INSCRIBE confident edit cluster center. **c**, **d** Actual fraction of motif-containing INSCRIBE edit clusters (dots) compared with that of the 20 permuted clusters (box plot, *n* = 20). Top: RBFOX2-INSCRIBE in PFA-fixed HEK293T cells. Z-scores: RBFOX2-INSCRIBE (PFA-fixed), 29.27; Enzyme-only control (PFA-fixed), 0.54; RBFOX2-STAMP: 23.08; Bottom: TDP-43-INSCRIBE in PFA-fixed HEK293T cells. Z-scores: TDP-43-INSCRIBE (PFA-fixed),

4.61; Enzyme-only control (PFA-fixed), −1.02. **d** The actual fraction of eCLIP-overlapping INSCRIBE edit clusters (diamonds) was significantly higher than that of the 20 permuted clusters (box plot, *n* = 20). Top: RBFOX2-INSCRIBE in PFA-fixed cells. Z-scores: RBFOX2-INSCRIBE (PFA-fixed), 18.67; Enzyme-only control (PFA-fixed), 1.36; RBFOX2-STAMP: 5.45; Bottom: TDP-43-INSCRIBE in PFA-fixed cells. Z-scores: TDP-43-INSCRIBE (PFA-fixed), 7.10; Enzyme-only control (PFA-fixed), −0.62. The actual fraction was plotted using the confident edit clusters derived from 3 INSCRIBE technical replicates. The box shows the quartiles while the whiskers extend to show the rest of the distribution; the median is represented by the center line. **e** Integrative genome viewer (IGV) tracks of example RBFOX2 target gene (*PKM*, *OGT*) with 1500 bp windows showing eCLIP peaks (orange), INSCRIBE in PFA-fixed cells (blue) and the enzyme-only control (gray). The overlay displays the edit fraction quantified by SAILOR in 3 replicates of INSCRIBE, alongside with the confident edit clusters identified by FLARE.

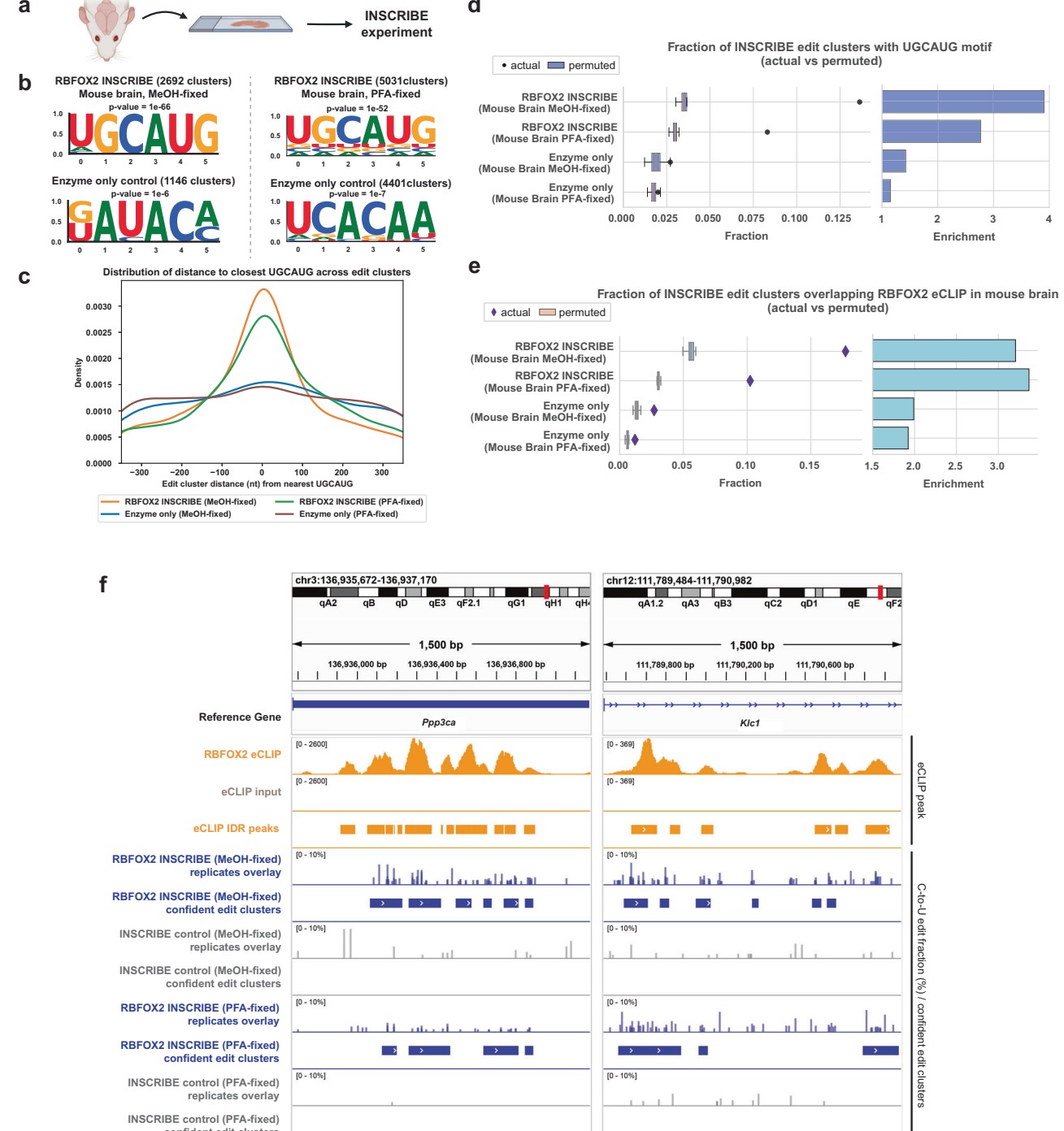

**Fig. 7 | INSCRIBE unveiled RNA targets of RBFOX2 in mouse brain tissue.**
**a** Schematics of RBFOX2-INSCRIBE with mouse brain tissue. Created with BioRender.com released under a Creative Commons Attribution-NonCommercial-NoDerivs 4.0 International license. **b** HOMER de novo motif discovery identified the canonical RBFOX2 motif UGCAUG as the top motif in MeOH-fixed and PFA-fixed mouse brain INSCRIBE confident edit clusters, but not the enzyme-only control, using a cumulative hypergeometric distribution for $p$ values. **c** A density plot of the distribution of the closest UGCAUG distance to confident edit clusters for INSCRIBE in MeOH-fixed mouse brain tissue (orange), PFA-fixed mouse brain tissue (green) and both enzyme-only controls (blue and purple). A peak at 0nt distance indicated the enrichment of UGCAUG in proximity to the edit cluster center. **d**, **e** The actual fraction of UGCAUG-containing mouse brain RBFOX2-INSCRIBE edit clusters (dots) was significantly higher than that of the 20 permuted clusters (box plot, $n = 20$). Z-scores: RBFOX2-INSCRIBE (Mouse Brain MeOH-fixed), 30.81; RBFOX2-INSCRIBE (Mouse Brain PFA-fixed), 28.05; Enzyme-only control (Mouse Brain MeOH-fixed), 1.95; Enzyme-only control (Mouse Brain PFA-fixed), 1.30. **e**, The actual fraction of mouse brain eCLIP-overlapping mouse brain RBFOX2-INSCRIBE edit clusters

(diamonds) was significantly higher than that of the 20 permuted clusters (box plot, $n = 20$) but not the enzyme-only control, demonstrating edit clusters specifically captured mouse RBFOX2 targeted RNA sites. The eCLIP experiment was performed in mouse whole brain and the IDR peaks were used for the comparison. Z-scores: RBFOX2-INSCRIBE (Mouse Brain MeOH-fixed), 37.75; RBFOX2-INSCRIBE (Mouse Brain PFA-fixed), 40.89; Enzyme-only control (Mouse Brain MeOH-fixed), 7.65; Enzyme-only control (Mouse Brain PFA-fixed), 6.06. In (**d**, **e**), the actual fraction was plotted using the confident edit clusters derived from 3 INSCRIBE technical replicates. Enrichment was defined as the ratio of actual fraction to the mean of the permuted-derived fraction. The box shows the quartiles while the whiskers extend to show the rest of the distribution; the median is represented by the center line. **f** Integrative genome viewer (IGV) tracks of example mouse RBFOX2 target genes (*Ppp3ca*, *Klc1*) with 1500 bp windows showing mouse RBFOX2 eCLIP peaks (orange), MeOH-fixed and PFA-fixed mouse brain RBFOX2-INSCRIBE (blue) and the respective enzyme-only controls (gray). The overlay displays the edit fraction quantified by SAILOR in 3 replicates of INSCRIBE, alongside with the confident edit clusters identified by FLARE.

## Animals

Two 2-month-old female C57BL/6 J wildtype mice were used in the study. Sex or age was not selected on purpose. The mouse strain was obtained from The Jackson Laboratory and bred and maintained in our laboratory. All mice were housed under specific pathogen-free conditions with a reversed 12-h light–dark cycle maintained at 23 °C with 30–70% humidity and provided with food and water *ad libitum*. The study was carried out in strict accordance with the recommendations in the Guide for the Care and Use of Laboratory Animals of the National Institutes of Health and approved by the Animal Care and Use Committee at University of California San Diego (Protocol Number: S12099).

## in situ RNA editing and RNA-seq library preparation

Approximately 150 K HEK293T cells were seeded on poly D-lysine (PDL, Sigma)-coated chamber of ibitreat 8-well slide (ibidi). Cells were cultured overnight to settle and attach to the slide.

After removing the medium and washing with PBS, the cells were fixed with 250 μL pre-chilled MeOH or 4% paraformaldehyde (PFA) for 15 mins on ice. For the PFA-fixed samples, cells were permeabilized with 0.3% Triton X-100 in PBS for 20 mins following fixation. All the washings and incubations were done on ice unless otherwise stated. The Washing Buffer consists of PBS with 2 mM DTT, 0.05% Tween-20. Following 3 cycles of washing in 300 μL Washing Buffer, cells were incubated with 100 μL blocking buffer (5% BSA in Washing Buffer with 0.2U/μL RNaseOUT) for 1 h. After blocking, the cells were incubated in 150 μL diluted primary antibodies (rabbit anti-RBFOX2, Bethyl, A300-864A; rabbit anti-TDP-43, Bethyl, A303-223A) with optimized concentration in Antibody Dilution Buffer (PBS with 1% BSA, 2 mM DTT and 0.2U/μL RNaseOUT) for 1 h. After 3 cycles of washing in PBS-T, 100 μL 1 μg/mL APO1-Nb in dilution buffer was added for 1-h incubation followed by 6 times of washing in PBS-T. The slides were then washed twice in 200 μL RNA editing buffer (20 mM Tris pH7.5, 20 mM NaCl, 5 mM DTT and 0.2U/μL RNaseOUT) on ice. Finally, the slides were incubated with 100uL RNA editing buffer at 37 °C for overnight (-16 h). For the Enzyme-only control, no antibody was added. For the Buffer-only control, no antibody or enzyme was added. Three replicates were performed in three separate slides for INSCRIBE experiments, enzyme-only controls and buffer-only controls.

On the next day, cells were harvested in 200 μL Trizol. Total RNA was extracted (Zymo Direct-zol RNA Purification Kit), followed by DNA depletion (DNase I treatment and Zymo RNA Clean and Concentrator Kit) and ribosomal RNA depletion (Illumina Ribo-Zero Plus rRNA Depletion Kit). For the PFA-fixed samples, cells were reverse-crosslinked prior to RNA extraction with 72 U/mL Proteinase K (NEB) in De-crosslinking Buffer (10 mM Tris pH8.0, 200 mM NaCl, 50 mM

EDTA, 2% SDS) for 2 h at 55 °C. The lysate post-decrosslinking were collected in Trizol LS. The RNA-seq library was prepared following the instruction of Illumina TruSeq Stranded Total RNA Library Preparation Kit. For the library preparation from ultra-low RNA input, 1 ng or 100 pg total RNA from each sample were used as starting materials in NEBNext® Single Cell/Low Input RNA Library Prep Kit (NEB) following the manufacturers' instructions.

For INSCRIBE in mouse brain tissue, the mouse brain was collected and snap frozen. The frozen brain was then cryosectioned into 20-μm slices and fixed in pre-chilled 100% methanol for 15 mins. It was then kept in methanol at −20 °C until INSCRIBE experiment. Each individual replicate of either experiment or control were prepared from one single slice of the brain (coronal section). After 3 cycles of washing in PBS-T, INSCRIBE was performed similarly as described above as fixed cells.

## Immunofluorescent imaging

Immunofluorescence (IF) was carried out similarly to the INSCRIBE protocol. All washings were done 3 times with PBS-T and the Antibody Dilution Buffer was used for all dilution series. For the primary antibody dilution series, after primary antibody (rabbit anti-RBFOX2 or rabbit anti-TDP-43) incubation and washing, the cells were incubated with 1:1000 goat anti-rabbit IgG Alexa 488 (Invitrogen, A11034) for 1 h, followed by DAPI stain and washing before being subjected to confocal imaging. For the APO1-Nb dilution series, after incubation with a fixed optimized concentration of primary antibody, cells were incubated with a series of APO1-Nb dilutions and then 1:1000 mouse anti-MBP antibodies (Proteintech, 66003-1-Ig). The cells were then washed and incubated with 1:1000 goat anti-rabbit IgG Alexa 488 and 1:1000 goat anti-mouse IgG Alexa 555 (Invitrogen, A21422). Subsequently, immunofluorescence samples were imaged on an LSM-880 fluorescence confocal microscope (Zeiss) equipped with 405 nm, 488 nm, and 561 nm lasers. Images were acquired with a 20x objective for traditional confocal microscopy and 63x for Airyscan superresolution imaging.

## Analysis of Immunofluorescence Images

Confocal images were processed and analyzed with Fiji (NIH). The "Threshold" function was used to determine the average intensity of antibody or nanobody staining for the corresponding fluorescence channel; the threshold value was set to capture all cellular intensity of the antibody or nanobody. To determine the fraction of nuclear signal in each image, we created masks of the whole cell using the antibody or nanobody channel and masks of the nucleus using the DAPI channel. The DAPI mask was subtracted from the whole-cell mask to create a cytoplasm-only mask of each image. The masks were subtracted from

the original image to remove extraneous intensity values, and the "Threshold" function was used to determine the intensity of the remaining pixels. The total intensity (i.e. area x intensity) of the nuclear antibody signal was divided by the total intensity of the whole cell to calculate the nuclear fraction. At least 400 cells were analyzed per condition. Colocalization of Airyscan images was calculated using the "Colocalization" analysis package in Fiji; the Pearson's correlation coefficient was determined for the nanobody and antibody channels of each image. Student's t-test was performed on the Pearson's correlation coefficient of 5 images to determine the p-value.

### Illumina sequencing and RNA mapping

RNA-seq libraries were sequenced with single-end reads (100 nucleotides) with a typical read depth of 60 M. Reads were then trimmed and filtered for repeat elements using sequences obtained from RepBase (v18.05) with STAR (2.4.0i). Reads that did not map to repeats were then mapped to the hg38 assembly with STAR (2.4.0i) for the HEK293T samples, or mapped to the mm10 assembly with STAR (2.5.2) for the mouse samples. The reads were then sorted with samtools (v1.9) and annotated against Gencode (v29).

### SAILOR quantification of RNA edits and FLARE identification of edit clusters

The resulting BAM files were each used as inputs to SAILOR (v1.1.0) to determine C > U edit sites across the hg38 assembly[12]. Briefly, SAILOR filters potential artifacts and known SNPs (dbSNP, v151) and returns a set of candidate edit sites and outputs the number of C > U conversions found among aligned reads. The C-to-U editing sites called with a confidence score larger than 0.5 and an editing percentage less than 0.8 were remained as the input to FLARE. The Poisson model-based FLARE accounts for background editing rates to filter false positives from truly edited regions, and scores identified clusters for use in downstream applications (https://github.com/YeoLab/FLARE). False discovery rate (FDR) threshold was set to 0.1 and the maximum merge distance was set to 15 nt. Clusters from three replicates were intersected through bedtools (2.27.1). [bedtools.intersect(wa=True, u=True, s=True)] Then, buffer-only clusters were subtracted from the intersected edit clusters and termed "confident edit clusters".

### PacBio long-read sequencing and identification of RNA edits

Iso-seq library preparation of RBFOX2-INSCRIBE and enzyme-only samples was completed using the Kinnex full-length RNA kit (PN 103-072-000) and associated protocol. Libraries were sequenced on the Pacbio Revio system, with one SMRT Cell allocated per sample. Following sequencing, the reads were processed using PacBio's software toolkit. Consensus circular sequence (CCS) reads were assembled using the ccs tool (v8.0.0) and standard parameters. Reads were then de-concatenated using skera (v1.2.0), trimmed using lima (v2.9.0) and refined using refine (v4.1.1). Full-length reads were aligned to the hg38 reference using pbmm2 (v1.13.0) with parameters: –preset ISOSEQ and –sort. Aligned reads were filtered for only primary mapped alignments using the SAM Flag. NanoPlot (v1.32.1)[30] was used with parameters: –raw and –tsv_stats. Reads with a quality score below 20, unmapped reads, supplementary alignment reads, secondary alignment reads, and reads aligned to the incorrect strand were excluded from the analysis. After read filtering, mRNA isoform-level counts matrices and read assignments were obtained using IsoQuant v3.3.0[31] with the following parameters: –data_type pacbio, –transcript_quantification unique_only, and –gene_quantification unique_only. Using custom scripts, cytosine-to-uracil edits were then called using the data. Putative SNPs were removed from the samples by identifying edits that occur across all replicates and samples. Remaining annotated SNPs

were removed using annotations from the dbSNP database[32]. The remaining edits were used to calculate gene and isoform-level editsC, the proportion of edited cytosines to all cytosines in a gene or isoform, respectively.

### INSCRIBE edit cluster evaluations

De novo motif discovery was performed using HOMER (v4.9.1)[19] using FLARE edit clusters and a shuffled background for each UTR, CDS, intron and total genic region. A kernel density estimation (KDE) plot was used to show the distribution of the closest motif (UGCAUG or UGUGUG) distance to confident edit clusters. Edit clusters were shuffled randomly within their respective exons or introns 20 times to derive the shuffled edit clusters. The STAMP experiments serving as comparisons to INSCRIBE were conducted in three replicates in HEK293T cells and computationally processed similarly. The eCLIP experiments compared to INSCRIBE were IDR (irreproducible discovery rate) peaks from RBFOX2 eCLIP in HEK293T cells (GSE77634), TDP-43 eCLIP in K562 cells (ENCODE Data Coordination Center, https://www.encodeproject.org) and RBFOX2 eCLIP in mouse whole brain tissue (GSE240326).

### Statistics & reproducibility

All INSCRIBE experiments were performed with three replicates. When "confident edit clusters" is indicated, only FLARE edit clusters that were consistent across all three replicates were considered for analysis. The experiments were not randomized. The Investigators were not blinded to allocation during experiments and outcome assessment.

### Reporting summary

Further information on research design is available in the Nature Portfolio Reporting Summary linked to this article.

## Data availability

All raw sequencing data and processed data for INSCRIBE in this study have been deposited in the Gene Expression Omnibus (GEO) database under accession code GSE240014, GSE240326, and GSE263371, respectively. The mouse brain eCLIP data used for comparison are available in the GEO database under accession code GSE240521. The previously published RBFOX2-STAMP data, RBFOX2 eCLIP data and TDP-43 eCLIP data used for comparison in this study are also available in the GEO database under accession code GSE232520, GSE155729 and GSE91895, respectively. Source Data file is provided with this paper. Imaging data in this study are available from the corresponding authors upon request, and requests will be fulfilled within 4 weeks. Source data are provided with this paper.

## Code availability

All custom software packages used for data analysis in this work, including SAILOR, and FLARE, is available at https://github.com/YeoLab/FLARE; other computational methods are described in Methods. Custom scripts are available at https://github.com/Lisa-LQS/INSCRIBE/[33].

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

## Acknowledgements

We thank Joshua Schwartz for assistance with mouse brain tissue, and Yan Song for providing eCLIP data for comparisons. We also thank the members in Yeo lab and Corbett lab for the support in preparing the manuscript, especially Karen Mei and Hema Kopalle. This work was supported by the Waitt Advanced Biophotonics Core Facility of the Salk Institute with funding from NIH-NCI CCSG: P30 014195 and the Waitt Foundation. K.R. was supported by the Milton Safenowitz Postdoctoral Fellowship from the ALS Association. Q.L. was supported by an individual predoctoral fellowship from the American Heart Association. The authors acknowledge support from the National Institutes of Health (R35 GM144121 to K.D.C., RF1 MH126719, R01 HG011864, R01 HG004659 to G.W.Y., and a subaward to G.W.Y. from U24 HG011735) and a Chan Zuckerberg Foundation Exploratory Cell Networks grant. This publication includes data generated at the UC San Diego IGM Genomics Center utilizing an Illumina NovaSeq 6000 that was purchased with funding from a National Institutes of Health SIG grant (#S10 OD026929) and data generated at the Sequencing Core Facility, La Jolla Institute utilizing an Illumina NovaSeq 6000 that was acquired through the Shared Instrumentation Grant (SIG) Program (#S10OD025052). Computational resources were provided by the Department of Defense High Performance Computing Modernization Program (HPCMP) and the Triton Shared Computing Cluster (TSCC) at the San Diego Supercomputer Center (SDSC)[34]. We would also like to thank the Stem Cell Genomics Core at the Sanford Stem Cell Institute for providing sequencing services.

## Author contributions

Q.L.- Conceptualization, Methodology, Software, Formal analysis, Investigation, Visualization, Writing—Original Draft, Review & Editing; T.Y.—Conceptualization, Methodology, Investigation, Writing—Review & Editing; E.K.- Software, Formal analysis; P.J.—Software, Formal analysis; K.R.—Investigation, Formal analysis; B.A.Y.—Software, Data Curation; K.D.C.—Resource, Writing—Review & Editing, Supervision; G.W.Y.—Resource, Writing—Review & Editing, Supervision.

## Competing interests

G.W.Y. is an SAB member of Jumpcode Genomics and a co-founder, member of the Board of Directors, on the SAB, equity holder, and paid consultant for Locanabio and Eclipse BioInnovations. G.W.Y. is a distinguished visiting professor at the National University of Singapore. G.W.Y.'s interests have been reviewed and approved by the University of California, San Diego in accordance with its conflict-of-interest policies. The authors declare no other competing financial or non-financial interests.
