## [Peer Review File · Nature Communications]

Reviewers' Comments:

Reviewer #1:

Remarks to the Author:

Summary

Liang and colleagues present their method INSCRIBE which combines features from antibody immunoprecipitation-based (CLIP) and editing-based methods (TRIBE and STAMP) to study RNA-RBP interactions. They use an APOBEC1-nanobody that targets a primary (rabbit) antibody against an RBP of interest, exemplified here by the extensively studied RBFOX2 and TDP-43.

They aim to address a major limiting methodological challenge: while the requirements for large amounts of input material have started to be addressed by the editing-based methods, to-date these approaches require expression of an exogenous RBP-enzyme. This limits their utility in important biological contexts where low input material is commonly an issue, for example clinical samples or spatial transcriptomics. Hence with INSCRIBE, the authors have developed an approach to identify RNA-RBP interactions through A-to-C editing, but via the endogenously expressed RBP. They demonstrate this in HEK293T cells and mouse brain tissue. This is an exciting method and strategy which has the potential to have wide-reaching applications.

However, I think there are two related areas which could benefit from further development to strengthen the manuscript and to enable the reader to appreciate the scope and limitations of INSCRIBE better, especially in comparison to well-understood CLIP methods. This is particularly important given the high degree of background noise seen even in the optimised higher input conditions ("noise elimination strategies... removed 97-98% of edit clusters in each individual replicate" - L165-168)

1. Throughout, the evidence that INSCRIBE recovers true binding sites is through overlap with eCLIP-derived peaks and through recovery of canonical RBFOX2 and TDP-43 binding motifs (UGCAUG and UGUGUG, respectively). In particular a lot of weight is given to the latter, but I think this warrants more extensive investigation.

2. Given that both RBFOX2 and TDP-43 are well-described splicing regulators an orthogonal assessment of how well INSCRIBE can recover regulatory binding patterns would be powerful and also help contextualise alongside CLIP.

Motifs and comparison with eCLIP peaks

Using enrichment as a metric and a permutational approach, the authors show recovery of the canonical binding motif. However, the absolute fractions show that the motif is present in between 8% (Fig 4B) to 10% (Fig 2E) of edit clusters, suggesting a large fraction of sites may reflect other binding motifs (Begg et al., 2020; Dominguez et al., 2018). For RBFOX2, half of CLIP peaks lack a GCAYG motif and it would be good to contextualise against this in the text (Begg et al., 2020).

Are other binding motifs recovered from the data, for example the alternative RBFOX2 motifs, or the UGAAUG TDP-43 motifs? This could either be through discussing the other motifs discovered by HOMER or for a more detailed analysis an approach such as mCross (Feng et al., 2019; Kuret et al., 2022) or PEKA may be useful to understand these alternative motifs and the positional relationship to editing clusters (as opposed to crosslink sites).

Are any motifs recovered from the "non-confident" or filtered out edit clusters? Could these give an insight into the potential technical biases/preferences of the INSCRIBE/APOBEC1 editing method? Or are potential "true" binding sites based on motifs also being filtered out?

Similar to the motifs, although there is enrichment of INSCRIBE edit clusters that overlap eCLIP

peaks, only a small fraction of clusters overlap (4% - Fig. 2F, 3% - Fig. 4B). I think this warrants a bit more exploration or explanation, for example: i) which eCLIP peaks are not recovered by INSCRIBE, ii) which INSCRIBE clusters are not detected with eCLIP (and are these detected by STAMP), iii) are there differences in the lengths/fragmentation of the INSCRIBE edit clusters compared with the eCLIP peaks, e.g. in Fig. 6F for *Ppp3ca* there are 6 clusters - although the eCLIP peaks are not shown (see also minor comment below) is this binding region called as 3-4 peaks, such that there are multiple INSCRIBE clusters per eCLIP peak or vice versa.

Some of the language may warrant toning down, for example when discussing the 100 pg total RNA work (Fig. 3D) in L240-243 although the binding sites were recovered in individual replicates, they were filtered out in the confident clusters. However, without the a priori knowledge from either eCLIP or 100 ng INSCRIBE it would not really be able to identify these accurately in isolation in a low-input/spatial transcriptomics scenario given the need for the stringent filtering to account for the background noise.

RNA maps

Using publicly-available shRNA-RBP knockdown (or CRISPR-KO) RNA-seq datasets to derive RNA maps would be a valuable orthogonal approach to show how INSCRIBE clusters recover splicing regulatory principles and help compare alongside with existing CLIP and STAMP methods to understand the sensitivity and specificity. The authors' lab has previously done this as part of ENCODE and extensively for RBFOX2 (e.g. (Yee et al., 2019)).

Long-read sequencing

The authors demonstrate that INSCRIBE is compatible with PacBio sequencing (Fig 2G), but this feels like a superficial exploration. It would be good to see some examples of differential binding/alternative binding patterns to different isoforms to support their statement that it "enables RNA isoform distinctions" (L190-191). A comment (or evidence) on whether it would work with other very commonly used long-read technologies, such as Oxford Nanopore (considering base-calling error rates) would be useful for future adopters of the method.

Other comments

It would be useful in a supplementary figure to see the degree of correlation between replicates of INSCRIBE (as is often done for CLIP) - perhaps most useful after removing the enzyme only background.

For the HOMER motif discovery (L515-6) was the input sequence precisely the edit cluster, or was it a window around the cluster as is often used? As the binding site/motif is often not the same as the crosslinking/editing site, using precisely the edit cluster may under-estimate the presence of a binding motif.

From e.g Fig 2D/3A/4C/5B, it appears that the binding motif is predominantly found in a region $\pm 50-100$ away from the edit cluster, which is quite a large range for an RBP and suggests a loss of spatial resolution. Would the authors be able to comment?

The authors note the potential biases that RNA structure may have on the recovery of binding sites. To understand this further, it may be useful to plot a metaprofile of pairing probability (e.g. obtained by RNA duplex from ViennaRNA or similar) around confident edit clusters. Comparing this with those around edit clusters that have been filtered out, or those from control conditions may be useful.

For the IGV screenshot figure panels where eCLIP signal and input tracks are shown, it would also be helpful to show the eCLIP IDR (or CLIPper/Skipper) peaks so they can also be compared with

the INSCRIBE edit clusters.

In all the figures the density plots are missing the y-axis values although the ticks are present.

Fig. 2B - The INSCRIBE confident edit cluster regions should be present in all 3 replicates, but the left-hand one (3'-most on the APP 3' UTR) does not appear to be in replicate 2, but only 1 and 3.

Typo on L 262: TARDP should be TARDBP

References

- Begg, B. E., Jens, M., Wang, P. Y., Minor, C. M., & Burge, C. B. (2020). Concentration-dependent splicing is enabled by Rbfox motifs of intermediate affinity. *Nature Structural & Molecular Biology*, 27(10), 901–912.
- Dominguez, D., Freese, P., Alexis, M. S., Su, A., Hochman, M., Palden, T., Bazile, C., Lambert, N. J., Van Nostrand, E. L., Pratt, G. A., Yeo, G. W., Graveley, B. R., & Burge, C. B. (2018). Sequence, Structure, and Context Preferences of Human RNA Binding Proteins. *Molecular Cell*, 70(5), 854–867.e9.
- Feng, H., Bao, S., Rahman, M. A., Weyn-Vanhentenryck, S. M., Khan, A., Wong, J., Shah, A., Flynn, E. D., Krainer, A. R., & Zhang, C. (2019). Modeling RNA-Binding Protein Specificity In Vivo by Precisely Registering Protein-RNA Crosslink Sites. *Molecular Cell*, 74(6), 1189–1204.e6.
- Kuret, K., Amalietti, A. G., Jones, D. M., Capitanichik, C., & Ule, J. (2022). Positional motif analysis reveals the extent of specificity of protein-RNA interactions observed by CLIP. *Genome Biology*, 23(1), 191.
- Yee, B. A., Pratt, G. A., Graveley, B. R., Van Nostrand, E. L., & Yeo, G. W. (2019). RBP-Maps enables robust generation of splicing regulatory maps. *RNA*, 25(2), 193–204.

Reviewer #2:

Remarks to the Author:

The development of methods allowing the identification of RNA-protein interactions in low sample amounts has the potential to be a real game changer in the RNA field and in studying RNA-protein interactions. High starting material requirements are not a trivial limitation when it comes to well established and widely accepted variations of the CLIP method, which is therefore not always usable for example precious patient samples. The INSCRIBE method presented by the authors here offers further improvement to already published methods like STAMP and TRIBE, allowing the study of RNA-protein interaction in samples without the need for exogenous expression of the RBP of interest fused to an RNA-editing enzymes. This firstly opens up the possibility of applying these low-input material requiring methodologies to a broader range of samples, like patient derived iPSCs, tissue samples and biopsies, which would be extremely interesting considering the disease relevance of RBPs. Secondly. Even in cellular context, exogenously expressing an RBP with another protein fused to it could affect the specificity of the RBP, whereas the method at hand circumvents this problem. Thirdly, it also improves on the existing CLIP methodologies, since it is applicable to fixed samples. I also find the manuscript well written, easy to follow and appreciate the work that has gone into this method development, especially considering the challenging task they are addressing. Although I have no doubt that people in the scientific community would find this a very useful tool, there are a few things, which I have outlined below that I think need to be addressed before publication.

Main comments:

With the way that the data is currently presented, it is a bit difficult to assess the specificity and reliability of the INSCRIBE method. If we take figure 2 as an example – here the authors show that around 10% of the reads overlap with the UGCAUG consensus motif of RBFOX2 and around 4% overlap with RBFOX2 eCLIP identified RBFOX2 binding sites. Although it is clear that this is much

higher than for the 'enzyme only' samples and seems to outperform STAMP, this still seems to be quite low and would require some further explanation to be convincing. I think the following aspects are very important to be addressed so that the data could be assessed better:

1. There needs to be a thorough discussion about the resolution of INSCRIBE. Since the APOBEC1 enzyme is bridged to the actual RBP by an antibody, it is clear that this distance will decrease resolution of the method compared to CLIP, where cross-linking allows for catching interaction at a so called 'zero distance'. It would be important to know what is the distance from the actual RNA-binding protein that APOBEC1 could modify the RNA at. The decreased resolution does of course not mean that the information gained is not valuable, however, it is crucial for people who will use this methodology in the future understand what the data they obtain means. Another question that arises, is whether this could also lead to modification of RNAs that are not directly bound by the RBP being investigated but are just found in the region?

2. I would also be good to further demonstrate if the clusters that are not overlapping with either CLIP sites or the consensus motif could be explained by the decreased resolution – i.e. is the INSCRIBE signal much more dispersed compared to the same site from the CLIP experiment? It would be good to present a graph similar to Figure 2D showing the distance of INSCRIBE clusters from RBFOX2 eCLIP sites. In figure 2 it is clear that there is an enrichment of confident INSCRIBE clusters around the consensus motif UGCAUG and I would like to see if that would be the case for eCLIP sites as well.

3. Another aspect is of course that even if an RBP has a defined consensus binding motif, its binding might not be limited to binding to that specific motif only. Therefore, I would like to ask the authors to add the RBFOX2 eCLIP data to the plot shown in figure 2B along with the INSCRIBE and STAMP data to demonstrate what is the fraction of eCLIP sites that contain the consensus motif. This would also be nice for Figure 4B for the TDP-43 INSCRIBE experiment, if the eCLIP data is available for such analysis.

4. I would further like to see the overlap of RBFOX2 target transcripts identified with eCLIP and INSCRIBE. This would allow to partially eliminate the problem of the lower resolution of the INSCRIBE method from the comparison of the two methods. That would tell us if the two methods identify largely the same target transcripts for RBFOX2? Is the overlap between the two datasets significant? Is the overlap between INSCRIBE data for RBFOX2 with the RBFOX2 eCLIP data bigger than the overlap between e.g. the RBFOX2 eCLIP data and the TDP-43 INSCRIBE data?

5. Lastly, it is important to add the enzyme only controls to the low-input and low sequencing depth figures. These are present in the other figures, but not figures 3 and supplementary figure 2. The authors claim that the method still works in these conditions, but in my opinion they need to show the relevant controls to make such a claim. For example, in SF2E, they show the enrichment of the RBFOX2 consensus sites for the 100 pg compared to the permuted clusters is slightly below 2-fold. In figure 6, they show that the enzyme only control shows also a 2-fold enrichment for the consensus motif compared to the permuted clusters. Therefore, as for the other experiments, it is important to also show here that the method outperforms the enzyme only control also for the 100 pg experiment.

The authors make a very exciting claim regarding the possibility of analyzing clinical and tissue samples and that their method allows analysis of samples fixed with different fixation agents. They proceed to demonstrate the applicability of INSCRIBE to methanol and paraformaldehyde fixed cells, but only demonstrate its applicability on methanol-fixed mouse tissue, which they themselves mention is preferential for RNA preservation. However, it is much more common for clinical tissue samples to be fixed in formaldehyde – which they also point out themselves in the results section where they analyze formaldehyde fixed cells – so it seems a bit strange that the authors stopped before they could fully demonstrate the applicability of INSCRIBE to these types of tissue samples. As it stands, it leaves the impression that the authors might've tried to apply

INSCRIBE to paraformaldehyde fixed tissue samples, but it did not work. I think it would be really important for the authors to either apply their method to tissue samples fixed in paraformaldehyde, or concede that testing whether this method will work on such tissue types will remain to be tested in future studies and tone down their claims regarding this.

Smaller comments:

1. In their introduction, they mention that CLIP requires high material input, due to loss during enrichment. I think in the interest of clarity, it would also be relevant to mention that it is also due to UV-cross-linking efficiency being very low (1%-5%), which is another issue that they circumvent in this method.

2. In figure 1, there is a figure legend for panel C, which does not seem to be present in the actual figure.

3. Since the authors have added additional data filtering steps to the previously published SAILOR and FLARE pipelines for identifying RNA editing events and clusters, it would be nice to somehow add this information to figure 1B, to make it clear that additional criteria have been used to ensure data quality.

4. Regarding the data filtering – the authors mention that they remove all edit clusters found in the enzyme only control. This is of course the most stringent way to approach this, but I am wondering if they would consider trying to retain clusters where they see enrichment in the INSCRIBE samples compared to the enzyme control. Since in the enzyme control samples, the APOBEC1 enzyme targets RNA randomly, it might still hit RBFOX2 targets, however, true RBFOX sites should still be enriched in the INSCRIBE samples.

5. Figure 2D is lacking units on the y-axis.

6. In the description for figure 2G, it is mentioned that the long read clusters overlap with the short read INSCRIBE clusters and eCLIP sites. I think for the sake of clarity, that sentence should reference to both figures 2G and 2B, since the statement is comparing the two figures.

7. In the method section describing in-situ RNA editing procedure, please add the amount of RNA-editing buffer added to the sample before over-night incubation. This would be nice to clarify, since this has been provided for the other steps and here it merely says that the samples are washed in RNA-editing buffer and then incubated at 37C overnight, which could be confusing.

8. For the PFA-fixed samples – could the authors please comment on how the fixation could affect the application of their method together with long-read sequencing, since I would expect the de-cross-linking step to only be partial and the remaining cross-links could interfere with the RT step.

Reviewer #3:

Remarks to the Author:

Authors report a novel technique to identify and study RNA-RNA binding protein interactions in cell culture and tissues. The so called IN situ Sensitive Capture of RNA-protein Interactions in Biological Environments (INSCRIBE) method uses APOBEC1-nanobody to bind to RBPs. As a proof-of-principle, authors were able to capture and identify two RBPs - RBFOX2 and TDP-43 in cells and tissue slices. The methodology used by authors is appropriate and the results obtained support their hypothesis and claims. INSCRIBE is an innovative technology which has good potential to study RNA-RBP interactions in various samples and is a better method compared to existing techniques with application in biomedical research. The manuscript can be accepted for publication after authors address the comments below;

-What was the rationale for selecting RBFOX2 and TDP-43 proteins for detection? There are several RBPs which carry binding motifs with different lengths. Both RBFOX2 and TDP-43 are able to recognize nucleotide sequence with six bases. What about RBPs that recognize longer nucleotide sequences? Wondering about the general applicability of this method to a wide range of mammalian RBPs

-What should be the ideal purity range of APOBEC1-nanobody to obtain consistent results

-Authors propose using this method to study RNA-RBP interactions in clinical/patient samples although no data is provided. Is it possible to use this technique to study RNA-RBP interactions in plasma samples? One important reference is missing (Cell Genom . 2023 Apr 20;3(5):100303. doi: 10.1016/j.xgen.2023.100303. eCollection 2023 May 10.)

-Figure 6: Not clear on how many brain slices were prepared per mouse and how many mice were used in the study? Why did the authors use female mice? These details should be clarified in the manuscript

-Minor comments: Author should use proper format while typing references in text. eg - page 3, paragraph 1, line 45 - "events1-3." should be "events.1-3". Punctuation "." should come before the reference numbers. Need to check for this throughout the manuscript

References section: Few references do not have proper journal format eg: 19, 22, 25 - words starting with upper case letters

Vol and page numbers are missing for some listed references

We thank the reviewers for their careful reading of the manuscript and their insightful comments and questions. We have submitted a copy of the revised manuscript with all substantive changes marked in **red text**. In addition, we addressed each reviewer's comments and questions below in **blue text**.

We'd like to remind the editor and the reviewers that this manuscript is a proof-of-concept of a novel technology that empowers the RBP-target discovery conveniently with low-input fixed cells or tissues, and opens the door to discover RBP functions in native biological contexts including clinical samples. We don't intend to perfect INSCRIBE in this manuscript that debuts the technology. Thus, in-depth analysis on motif discovery and splicing regulatory pattern discovery are beyond the scope of this study, and are not included in the manuscript. We plan to further advance the technology and our understanding of INSCRIBE in a future manuscript.

The highlights of the major changes in the manuscript are listed below:

1. A more intensive analysis of the long-read RBFOX2-INSCRIBE has been integrated to the new **Figure 3**, showcasing INSCRIBE's capability to distinguish RNA isoform-specific RBP binding.
2. New data of RBFOX2-INSCRIBE in PFA-fixed mouse brain have been appended to the new **Figure 7** along with the previous data in MeOH-fixed mouse brain, demonstrating INSCRIBE's great potential to address formaldehyde-fixed clinical samples.
3. A more comprehensive examination of the INSCRIBE technology has been included in the new **Discussion** section with the new **Supplementary Figure 8**, addressing (1) how INSCRIBE compares to eCLIP; (2) what's INSCRIBE's RBP target detection resolution; (3) accuracy of INSCRIBE motif discovery.

Responses to Reviewer #1:

Summary

Liang and colleagues present their method INSCRIBE which combines features from antibody immunoprecipitation-based (CLIP) and editing-based methods (TRIBE and STAMP) to study RNA-RBP interactions. They use an APOBEC1-nanobody that targets a primary (rabbit) antibody against an RBP of interest, exemplified here by the extensively studied RBFOX2 and TDP-43.

They aim to address a major limiting methodological challenge: while the requirements for large amounts of input material have started to be addressed by the editing-based methods, to-date these approaches require expression of an exogenous RBP-enzyme. This limits their utility in important biological contexts where low input material is commonly an issue, for example clinical samples or spatial transcriptomics. Hence with INSCRIBE, the authors have developed an approach to identify RNA-RBP interactions through A-to-C editing, but via the endogenously expressed RBP. They demonstrate this in HEK293T cells and mouse brain tissue. This is an exciting method and strategy which has the potential to have wide-reaching applications.

However, I think there are two related areas which could benefit from further development to strengthen the manuscript and to enable the reader to appreciate the scope and limitations of INSCRIBE better, especially in comparison to well-understood CLIP methods. This is particularly important given the high degree of background noise seen even in the optimised higher input conditions (“noise elimination strategies... removed 97-98% of edit clusters in each individual replicate” - L165-168)

1. Throughout, the evidence that INSCRIBE recovers true binding sites is through overlap with eCLIP-derived peaks and through recovery of canonical RBFOX2 and TDP-43 binding motifs (UGCAUG and UGUGUG, respectively). In particular a lot of weight is given to the latter, but I think this warrants more extensive investigation.

More extensive discussions about motifs are included in our responses below and also added to the manuscript “Discussion” section.

2. Given that both RBFOX2 and TDP-43 are well-described splicing regulators an orthogonal assessment of how well INSCRIBE can recover regulatory binding patterns would be powerful and also help contextualise alongside CLIP.

This is a very good suggestion. Please see the discussion under the “RNA maps” section below about the analysis on splicing regulatory binding patterns.

Motifs and comparison with eCLIP peaks

Using enrichment as a metric and a permutational approach, the authors show recovery of the canonical binding motif. However, the absolute fractions show that the motif is present in between 8% (Fig 4B) to 10% (Fig 2E) of edit clusters, suggesting a large fraction of sites may

reflect other binding motifs (Begg et al., 2020; Dominguez et al., 2018). For RBFOX2, half of CLIP peaks lack a GCAYG motif and it would be good to contextualise against this in the text (Begg et al., 2020).

We agreed that the apparent low fraction of motif presence could be due to the promiscuous binding nature of the RBP, and not all RBP binding sites reflect the canonical motif, as in the literature quoted by the reviewer. We added the discussion in the revised manuscript.

Are other binding motifs recovered from the data, for example the alternative RBFOX2 motifs, or the UGAAUG TDP-43 motifs? This could either be through discussing the other motifs discovered by HOMER or for a more detailed analysis an approach such as mCross (Feng et al., 2019; Kuret et al., 2022) or PEKA may be useful to understand these alternative motifs and the positional relationship to editing clusters (as opposed to crosslink sites).

We thank the reviewer for the excellent suggestions on investigating alternative motifs. We first examined the HOMER discovered motifs from the confident edit clusters, and we observed a significantly larger P-value for other 6-mer motifs (P-value = 1.0e-11 for the 2nd top motif), compared to “UGCAUG” for RBFOX2 (P-value = 1.0e-154). This indicates HOMER did not discover a strong alternative RBFOX2 motif from the data. For TDP-43, we plotted density plots for all 6-mers that the 2nd top HOMER motif (shown as the motif image below) could represent, and found only minor enrichment for “AUGACU”. We didn’t see enrichment for “UGAAUG” (the other mCross motif) particularly, but we did see a higher enrichment for “UGNNUG” compared to the enzyme only control. Among those, “UGUGUG”, “UGUAUG”, “UGUCUG”, “UGCGUG” and “UGAGUG” (termed “alternative UGNNUG motifs” in the below figure) contribute the most to “UGNNUG” enrichment, suggesting them to be the alternative motifs that we missed.

We explored whether the alternative/secondary RBF0X2 motifs are enriched in the INSCRIBE clusters that do not overlap with eCLIP. Referring to a similar analysis in FLARE paper (Kofman et al., 2023), we plotted the fraction of INSCRIBE clusters containing GCAUG as well as 7 top secondary motifs found by Begg et al.: GCACG, GCUUG, GAAUG, GUUUG, GUAUG, GUGUG and GCCUG. Compared to a set of control motifs, the fraction of eCLIP-overlapping clusters containing GCAUG + secondary motifs is of a similar magnitude to that of eCLIP-overlapping clusters (panel A below; Mann–Whitney U test p-value: eCLIP-overlapping clusters-0.011; eCLIP-non-overlapping clusters-0.090), indicating that many of these INSCRIBE clusters may reflect real binding events at secondary motifs, as opposed to the Enzyme only INSCRIBE clusters (panel B below; eCLIP-overlapping cluster p-value=0.245, eCLIP-non-overlapping cluster p-value=0.428).

Are any motifs recovered from the “non-confident” or filtered out edit clusters? Could these give an insight into the potential technical biases/preferences of the INSCRIBE/APOBEC1 editing method? Or are potential “true” binding sites based on motifs also being filtered out?

Inspired by the above comment, we explored the filtered-out edit clusters in more depth. We found some level of enrichment of “UGCAUG” motif in the “filtered-out” edit clusters, albeit to a much lesser extent than the enrichment of confident edit clusters (see the below figure). This indicates that we indeed filtered out potential “true” binding sites. We have been striving for an effective pipeline to filter out noise while retaining true binding sites (the FLARE software from our lab is one of the efforts); however, we acknowledge that the current pipeline plus the straightforward 3-replicate intersection strategy has room for improvement.

Similar to the motifs, although there is enrichment of INSCRIBE edit clusters that overlap eCLIP peaks, only a small fraction of clusters overlap (4% - Fig. 2F, 3% - Fig. 4B). I think this warrants a bit more exploration or explanation, for example: i) which eCLIP peaks are not recovered by INSCRIBE, ii) which INSCRIBE clusters are not detected with eCLIP (and are these detected by STAMP), iii) are there differences in the lengths/fragmentation of the INSCRIBE edit clusters compared with the eCLIP peaks, e.g. in Fig. 6F for Ppp3ca there are 6 clusters - although the eCLIP peaks are not shown (see also minor comment below) is this binding region called as 3-4 peaks, such that there are multiple INSCRIBE clusters per eCLIP peak or vice versa.

We appreciate the very insightful comments from the reviewer. This reflects the nature of the INSCRIBE experiment, where APOBEC1 does not edit at the actual binding site because (1) the immediate sites occupied by RBFOX2 are not accessible for editing (2) APOBEC1 recognizes the RBP through a fairly long antibody-nanobody linkage, as mentioned in FLARE (Kofman et al., 2023).

i) ii) To further investigate what RBP binding sites each method discovered, we plotted the density plot of closest UGCAUG distance for eCLIP, INSCRIBE, overlapped, and non-overlapped windows (See figure below). Whether it was eCLIP, INSCRIBE, overlapping windows of two, or non-overlapping windows of two, there was a significant enrichment of UGCAUG near the center of the binding sites, indicating both experiments captured true but separate sets of binding sites. We then asked what was unique in the two sets of binding sites and thus what caused the low overlapping rate between these experiments. By plotting the regions of the discovered binding sites of eCLIP, INSCRIBE, overlapping windows of two, or the non-overlapping windows of two, we found that the two methods revealed distinct profiles of RBP target regions. INSCRIBE has much higher signals in 3'-UTR but less intronic regions as discussed in the Discussion section of the manuscript, compared to eCLIP. The low overlapped ratio of the two experiments is partially due to INSCRIBE's caveat in detecting intronic binding sites. On the other hand, this analysis also suggests that both methods have their own bias and INSCRIBE could identify novel targets.

A**B**
We further similarly compared INSCRIBE and STAMP. Although both methods reflected true RBFOX2 binding events (see figure below), the profile of regions of RNA target in the overlapped and non-overlapped windows were highly dependent on the methodology itself. STAMP has much fewer sites detected in the intronic area, which affects the overlapping of detected binding windows between the two experiments greatly. In addition, STAMP relied on the over-expression of the RBP-APOBEC1 fusion protein and editing over a period of time in live cells while INSCRIBE depicted a snapshot of RBP binding in a more native RBP expression level. Thus, it is not surprising that they reveal different binding sites and the overlapping ratio is low.

A**B**
iii) This is a great point. Inspired by the comment, we plotted the length distribution of eCLIP peaks vs INSCRIBE edit clusters. INSCRIBE cluster lengths were a series of discrete values due to the fixed-window-based edit sites merging strategy in FLARE. Most INSCRIBE clusters had a length of 45bp; some other typical lengths are 75bp, 90bp and 105bp. In contrast, eCLIP peaks, which represent the RBP cross-linking sites and reflect the accessibility of the RNAs bound by the RBP, have continuous peak lengths. The eCLIP peaks showed a larger mean length (68.27bp) compared to INSCRIBE clusters (59.91bp) and the length distribution was more dispersed than INSCRIBE clusters. In terms of the peak/cluster distance, INSCRIBE clusters are

more spaced out than eCLIP peaks (among all the peaks within a 10kb distance). Multiple INSCRIBE clusters can overlap with one eCLIP peak but that should not be the majority case, since there were very few INSCRIBE clusters located <100bp to each other.

Some of the language may warrant toning down, for example when discussing the 100 pg total RNA work (Fig. 3D) in L240-243 although the binding sites were recovered in individual replicates, they were filtered out in the confident clusters. However, without the a priori knowledge from either eCLIP or 100 ng INSCRIBE it would not really be able to identify these accurately in isolation in a low-input/spatial transcriptomics scenario given the need for the stringent filtering to account for the background noise.

We appreciate the reviewer pointing this out. Indeed, practically, it's hard to confidently identify a binding location if the confident cluster (the overlap of three replicates) presents at that location, even though all three individual replicates were recovered in the nearby loci. We admit that with lower input RNA and thus lower coverage per transcript, we might miss out on some RBP targets. The limitation of low-input INSCRIBE is reflected by the number of confident clusters we identified in the low-input samples. The limitation can potentially be fixed by pseudo-bulk analysis

moving towards spatial transcriptomics INSCRIBE. We revised the main text to address more on the limitation of low-input INSCRIBE.

RNA maps

Using publicly-available shRNA-RBP knockdown (or CRISPR-KO) RNA-seq datasets to derive RNA maps would be a valuable orthogonal approach to show how INSCRIBE clusters recover splicing regulatory principles and help compare alongside with existing CLIP and STAMP methods to understand the sensitivity and specificity. The authors' lab has previously done this as part of ENCODE and extensively for RBFOX2 (e.g. (Yee et al., 2019)).

We thank the reviewer for the great suggestion. In order to investigate whether INSCRIBE recovers splicing regulatory principles, we used a similar splicing analysis in the more recent antibody-barcode eCLIP (ABC) publication (Lorenz et al., 2023). By chi-squared test, upstream intron and exon cassette binding of RBFOX2 could explain exon inclusion events while the RBFOX2 binding on downstream introns could explain exon exclusion events (see figure below). For the eCLIP IDR peaks, there's enrichment for RBFOX2 binding upstream to the included exons in the mature mRNA. There's also enrichment for RBFOX2 binding downstream to the excluded exons in the mature mRNA.

RBFOX2 INSCRIBE in both MeOH-fixed and PFA-fixed condition (~60M reads) did not show enriched RBFOX2 binding at those regions. This is likely due to the fact that INSCRIBE could not capture enough intronic binding sites because of the low coverage of intronic reads. This has also been explored in the "low sequencing depth INSCRIBE" section. Since increasing the sequencing depth could improve the discovery of intronic edit clusters, we repeated the splicing analysis on the 100M read-depth RBFOX2 INSCRIBE data, which indeed showed an enrichment of RBFOX2 binding downstream of the excluded exons and demonstrated INSCRIBE could detect true binding sites with more read depths. We concluded that at this stage, INSCRIBE can provide limited splicing regulatory information for the RBP-of-interest transcriptome-wide and one can increase sequencing depth or couple INSCRIBE with targeted sequencing to extract splicing regulatory information.

Long-read sequencing

The authors demonstrate that INSCRIBE is compatible with PacBio sequencing (Fig 2G), but this feels like a superficial exploration. It would be good to see some examples of differential binding/alternative binding patterns to different isoforms to support their statement that it “enables RNA isoform distinctions” (L190-191). A comment (or evidence) on whether it would work with other very commonly used long-read technologies, such as Oxford Nanopore (considering base-calling error rates) would be useful for future adopters of the method.

We thank the reviewer for commenting on the long-read sequencing. We added a new batch of long-read sequencing data from Pacbio in the manuscript (see Figure 3), where we showed INSCRIBE editing on different RNA isoforms. Due to timeline constraints, we were not able to assess other long-read platforms such as Oxford Nanopore. However, we believe it is compatible with INSCRIBE, since the similar technology STAMP demonstrated the compatibility with both Pacbio and Oxford Nanopore (Brannan et al., 2021). We mentioned this in the revised manuscript.

Other comments

It would be useful in a supplementary figure to see the degree of correlation between replicates of INSCRIBE (as is often done for CLIP) - perhaps most useful after removing the enzyme only background.

We agree that plotting a degree of correlation between replicates would be helpful. We plotted the correlation of edit fractions across replicates in transcriptomic regions of RBFOX2-INSCRIBE edit clusters and calculated the Pearson correlations ($R=0.83$ and 0.81 for rep1 vs rep2 and rep2 vs rep3, respectively). The following figure is now added to Supplementary Figure 2.

INSCRIBE edit fraction reproducibility

For the HOMER motif discovery (L515-6) was the input sequence precisely the edit cluster, or was it a window around the cluster as is often used? As the binding site/motif is often not the same as the crosslinking/editing site, using precisely the edit cluster may under-estimate the presence of a binding motif.

We only used the edit cluster itself for HOMER motif discovery, without extending the window. It is true that the edit cluster might not contain the binding motif (i.e. the precise binding site) due to the relatively lowered resolution derived from the INSCRIBE/STAMP design (discussed below). The fact that HOMER motif discovery still showed strong enrichment of UGCAUG for only the cluster region further emphasized the strong specificity of INSCRIBE.

From e.g Fig 2D/3A/4C/5B, it appears that the binding motif is predominantly found in a region ± 50 -100 away from the edit cluster, which is quite a large range for an RBP and suggests a loss of spatial resolution. Would the authors be able to comment?

We appreciate the reviewer for bringing this up. This is due to the nature of the INSCRIBE experiment, in which the APOBEC1 does not edit at the precise binding site occupied by the RBP. It indeed compromised the spatial resolution of the RBP-binding sites detection. To assess the resolution of INSCRIBE, we plotted the cumulative distribution of the distance between the INSCRIBE edit clusters and the nearest UGCAUG motif. For RBFOX2 INSCRIBE, in both MeOH-fixation and PFA-fixation conditions, the normalized edit count within a fixed distance to UGCAUG increases as the distance increases, albeit at a different speed (slope). The “Enzyme only” curve reflected the background APOBEC1 editing relative to the UGCAUG distribution in the transcriptome and the slope of the curve remained constant; in contrast, the INSCRIBE curve had a much higher slope in the first ~ 50 nt, and then converge with the “Enzyme only” at ~ 200 nt, indicating the edits within ~ 200 nt were driven by RBFOX2 binding. We thus concluded the resolution of APOBEC1-nanobody editing in the INSCRIBE experiment is about 200nt. We have now added it in Supplementary Figure 8 and the Discussion section.

Cumulative distribution of INSCRIBE edit clusters distance to UGCAUG

Cumulative distribution of INSCRIBE edit clusters distance to UGCAUG

Slope of the cumulative distribution curve

Slope of the cumulative distribution curve

The authors note the potential biases that RNA structure may have on the recovery of binding sites. To understand this further, it may be useful to plot a metaprofile of pairing probability (e.g. obtained by RNA duplex from ViennaRNA or similar) around confident edit clusters. Comparing this with those around edit clusters that have been filtered out, or those from control conditions may be useful.

We appreciate the reviewer's suggestion to evaluate the edit bias derived from RNA structures. We calculated the minimal free energy (MFE) of the sequences in confident clusters as well as the filtered-out clusters from each replicate in RBFOX2 INSCRIBE. As shown in the figure below, there's no significant difference between MFE distribution of the confident edit clusters sequences and the filtered-out sequences. We suspect that the overall secondary and tertiary RNA structures may affect the enzymatic reaction by affecting the substrate cytosine accessibility, however, these might be beyond what the RNA folding programs can distinguish. Other than the folded RNA structures, a more crucial parameter affecting the enzyme kinetics is the local base context upstream and downstream of the editable "C". We and others have found the context preference of the edited "C" for a variety of APOBEC family enzymes. As far as we noticed, APOBEC1 tends to have higher enzymatic activity on a substrate "C" with A/T at its 5'-end (Medina-Munoz et al., 2024).

For the IGV screenshot figure panels where eCLIP signal and input tracks are shown, it would also be helpful to show the eCLIP IDR (or CLIPper/Skipper) peaks so they can also be compared with the INSCRIBE edit clusters.

We agree with the reviewer and we have added the eCLIP IDR peak on all the IGV screenshots.

In all the figures the density plots are missing the y-axis values although the ticks are present.

We thank the reviewer for pointing out that the y-axis is missing. We have updated the figures with the correct axis label.

Fig. 2B - The INSCRIBE confident edit cluster regions should be present in all 3 replicates, but the left-hand one (3'-most on the APP 3' UTR) does not appear to be in replicate 2, but only 1 and 3.

The edit cluster intersection among 3 replicates was carried out using the “bedtools intersect”. Replicate 1 (file A) and replicate 2 (file B) were intersected first and output the intersected region of replicate 1 (file A); subsequently, the resulting intersected region represented by file A region was intersected with replicate 3 (file C). (Source code: `bedtools.intersect(wa=True, u=True, s=True)`, where “wa” means “write the original entry in A for each overlap”)

Typo on L 262: TARDP should be TARDBP

We apologize for the oversight. We have now fixed it in the manuscript.

Responses to Reviewer #2:

The development of methods allowing the identification of RNA-protein interactions in low sample amounts has the potential to be a real game changer in the RNA field and in studying RNA-protein interactions. High starting material requirements are not a trivial limitation when it comes to well established and widely accepted variations of the CLIP method, which is therefore not always usable for example precious patient samples. The INSCRIBE method presented by the authors here offers further improvement to already published methods like STAMP and TRIBE, allowing the study of RNA-protein interaction in samples without the need for exogenous expression of the RBP of interest fused to an RNA-editing enzymes. This firstly opens up the possibility of applying these low-input material requiring methodologies to a broader range of samples, like patient derived iPSCs, tissue samples and biopsies, which would be extremely interesting considering the disease relevance of RBPs. Secondly. Even in cellular context, exogenously expressing an RBP with another protein fused to it could affect the specificity of the RBP, whereas the method at hand circumvents this problem. Thirdly, it also improves on the existing CLIP methodologies, since it is applicable to fixed samples. I also find the manuscript well written, easy to follow and appreciate the work that has gone into this method development, especially considering the challenging task they are addressing. Although I have no doubt that people in the scientific community would find this a very useful tool, there are a few thing, which I have outlined below that I think need to be addressed before publication.

Main comments:

With the way that the data is currently presented, it is a bit difficult to assess the specificity and reliability of the INSCRIBE method. If we take figure 2 as an example – here the authors show that around 10% of the reads overlap with the UGCAUG consensus motif of RBFOX2 and around 4% overlap with RBFOX2 eCLIP identified RBFOX2 binding sites. Although it is clear that this is much higher than for the ‘enzyme only’ samples and seems to outperform STAMP, this still seems to be quite low and would require some further explanation to be convincing. I think the following aspects are very important to be addressed so that the data could be assessed better:

1. There needs to be a thorough discussion about the resolution of INSCRIBE. Since the APOBEC1 enzyme is bridged to the actual RBP by an antibody, it is clear that this distance will decrease resolution of the method compared to CLIP, where cross-linking allows for catching interaction at a so called ‘zero distance’. It would be important to know what is the distance from the actual RNA-binding protein that APOBEC1 could modify the RNA at. The decreased resolution does of course not mean that the information gained is not valuable, however, it is crucial for people who will use this methodology in the future understand what the data they obtain means. Another question that arises, is whether this could also lead to modification of RNAs that are not directly bound by the RBP being investigated but are just found in the region?

We appreciate the very insightful comments from the reviewer. This is due to the nature of the INSCRIBE experiment, in which the APOBEC1 does not edit at the precise binding site occupied by the RBP. It indeed compromised the spatial resolution of the RBP-binding sites detection. To

assess the resolution of INSCRIBE, we plotted the cumulative distribution of the distance between the INSCRIBE edit clusters and the nearest UGCAUG motif. For RBFOX2 INSCRIBE, in both MeOH-fixation and PFA-fixation conditions, the normalized edit count within a fixed distance to UGCAUG increases as the distance increases, albeit at a different speed (slope). The “Enzyme only” curve reflected the background APOBEC1 editing relative to the UGCAUG distribution in the transcriptome and the slope of the curve remained constant; in contrast, the INSCRIBE curve had a much higher slope in the first ~50nt, and then converge with the “Enzyme only” at ~200nt, indicating the edits within ~200nt were driven by RBFOX2 binding. We thus concluded the resolution of APOBEC1-nanobody editing in the INSCRIBE experiment is about 200nt. We have now added it in Supplementary Figure 8 and the Discussion section.

2. I would also be good to further demonstrate if the clusters that are not overlapping with either CLIP sites or the consensus motif could be explained by the decreased resolution – i.e. is the INSCRIBE signal much more dispersed compared to the same site from the CLIP experiment? It would be good to present a graph similar to Figure 2D showing the distance of INSCRIBE clusters from RBFOX eCLIP sites. In figure 2 it is clear that there is an enrichment of confident INSCRIBE clusters around the consensus motif UGCAUG and I would like to see if that would be the case for eCLIP sites as well.

This is a very good point. Inspired by this comment, we examined the distribution of the distances between INSCRIBE clusters, between eCLIP peaks, and between INSCRIBE-eCLIP (under

10kb). As the left figure below shows, INSCRIBE signals were much more dispersed than the eCLIP experiment, which at least partially explains the low overlap with eCLIP.

We also plotted the distribution of the distances between INSCRIBE clusters and the nearest eCLIP peaks and showed a clear enrichment for the RBFOX2 INSCRIBE vs Enzyme only, as the right figure below shows. This figure has been included in the manuscript (Figure 2).

3. Another aspect is of course that even if an RBP has a defined consensus binding motif, its binding might not be limited to binding to that specific motif only. Therefore, I would like to ask the authors to add the RBFOX2 eCLIP data to the plot shown in figure 2B along with the INSCRIBE and STAMP data to demonstrate what is the fraction of eCLIP sites that contain the consensus motif. This would also be nice for Figure 4B for the TDP-43 INSCRIBIE experiment, if the eCLIP data is available for such analysis.

We agree with the reviewer on adding eCLIP data for comparison. We have added RBFOX2 eCLIP data in Fig.2B and TDP-43 eCLIP data in Fig.4B.

4. I would further like to see the overlap of RBFOX2 target transcripts identified with eCLIP and INSCRIBE. This would allow to partially eliminate the problem of the lower resolution of the INSCRIBE method from the comparison of the two methods. That would tell us if the two methods identify largely the same target transcripts for RBFOX2? Is the overlap between the two datasets significant? Is the overlap between INSCRIBE data for RBFOX2 with the RBFOX2 eCLIP data bigger than the overlap between e.g. the RBFOX2 eCLIP data and the TDP-43 INSCRIBE data?

Inspired by this great suggestion, we plotted the overlapping ratio at transcript level intersecting INSCRIBE and eCLIP gene targets. First, in comparison with RBFOX2 eCLIP (IDR peaks), RBFOX2, Enzyme only or TDP-43 INSCRIBE (confident edit clusters) each showed a similar fraction of transcript overlap. Even the TDP43 eCLIP (IDR peaks) showed a similar fraction of overlap. Note that all the files compared were comparable in total gene number (ranging from 1713 genes to 3336 genes), the overlapping factions should not be much affected by the varied size of different gene sets. Second, in a similar analysis comparing experiments with TDP-43

eCLIP, all the overlapping transcript fractions are similar including RBFOX2 eCLIP. We thus deem this analysis non-conclusive. We believe that both RBFOX2 and TDP43 are RBPs with a very broad range of targets, and therefore, the gene-level analysis is not the best to evaluate INSCRIBE specificity.

5. Lastly, it is important to add the enzyme only controls to the low-input and low sequencing depth figures. These are present in the other figures, but not figures 3 and supplementary figure 2. The authors claim that the method still works in these conditions, but in my opinion they need to show the relevant controls to make such a claim. For example, in SF2E, they show the enrichment of the RBFOX2 consensus sites for the 100 pg compared to the permuted clusters is slightly below 2-fold. In figure 6, they show that the enzyme only control shows also a 2-fold enrichment for the consensus motif compared to the permuted clusters. Therefore, as for the other experiments, it is important to also show here that the method outperforms the enzyme only control also for the 100 pg experiment.

We agree with the reviewer on this matter. All the IGV tracks of the enzyme-only controls have now been added to Supplementary Figure 5. The enzyme-only data have also been added to Supplementary Figure 4B, C, E, F.

The authors make a very exciting claim regarding the possibility of analyzing clinical and tissue samples and that their method allows analysis of samples fixed with different fixation agents. They proceed to demonstrate the applicability of INSCRIBE to methanol and paraformaldehyde fixed cells, but only demonstrate its applicability on methanol-fixed mouse tissue, which they themselves mention is preferential for RNA preservation. However, it is much more common for clinical tissue samples to be fixed in formaldehyde – which they also point out themselves in the results section where they analyze formaldehyde fixed cells – so it seems a bit strange that the

authors stopped before they could fully demonstrate the applicability of INSCRIBE to these types of tissue samples. As it stands, it leaves the impression that the authors might've tried to apply INSCRIBE to paraformaldehyde fixed tissue samples, but it did not work. I think it would be really important for the authors to either apply their method to tissue samples fixed in paraformaldehyde, or concede that testing whether this method will work on such tissue types will remain to be tested in future studies and tone down their claims regarding this.

We appreciate the comments about INSCRIBE in PFA-fixed mouse brain tissue. We have conducted the experiment and added RBFOX2-INSCRIBE results in PFA-fixed mouse brain tissue in Figure 7, demonstrating the applicability of INSCRIBE in PFA-fixed clinical tissue samples.

Smaller comments:

1. In their introduction, they mention that CLIP requires high material input, due to loss during enrichment. I think in the interest of clarity, it would also be relevant to mention that it is also due to UV-cross-linking efficiency being very low (1%-5%), which is another issue that they circumvent in this method.

This is a very good point and we have added this to the manuscript.

2. In figure 1, there is a figure legend for panel C, which does not seem to be present in the actual figure.

We thank the reviewer for bringing this to our attention and apologize for our error. This has been fixed in the manuscript.

3. Since the authors have added additional data filtering steps to the previously published SAILOR and FLARE pipelines for identifying RNA editing events and clusters, it would be nice to somehow add this information to figure 1B, to make it clear that additional criteria have been used to ensure data quality.

We appreciate the great suggestion for the clarity of the manuscript. We have added it to Figure 1B.

4. Regarding the data filtering – the authors mention that they remove all edit clusters found in the enzyme only control. This is of course the most stringent way to approach this, but I am wondering if they would consider trying to retain clusters where they see enrichment in the INSCRIBE samples compared to the enzyme control. Since in the enzyme control samples, the APOBEC1 enzyme targets RNA randomly, it might still hit RBFOX2 targets, however, true RBFOX sites should still be enriched in the INSCRIBE samples.

We explored the filtered-out edit clusters in more depth. We found some enrichment of “UGCAUG” motif in the “filtered-out” edit clusters, albeit to a much lesser extent than the enrichment of confident edit clusters. This indicates that we indeed filtered out potential “true” binding sites but to a much lesser proportion compared to the noise being removed. We have

been striving for an effective pipeline to filter out noise while retaining true binding sites (the FLARE software from our lab is one of the efforts); however, we acknowledge that the current pipeline plus the straightforward 3-replicate intersection strategy has room for improvement. We chose to maintain a high stringency but we also leave this open to the readers if they would prefer to trade off for a better sensitivity.

5. Figure 2D is lacking units on the y-axis.

We thank the reviewer for pointing this out. We have updated the figure with the correct axis label.

6. In the description for figure 2G, it is mentioned that the long read clusters overlap with the short read INSCRIBE clusters and eCLIP sites. I think for the sake of clarity, that sentence should reference to both figures 2G and 2B, since the statement is comparing the two figures.

We thank the reviewer for the detailed and careful examination of the manuscript. We have generated a more comprehensive batch of long-read sequencing data and have added the results to the revised figure (Figure 3). We have made sure to refer to all relevant figures properly in the main text for the new long-read sequencing figure.

7. In the method section describing in-situ RNA editing procedure, please add the amount of RNA-editing buffer added to the sample before over-night incubation. This would be nice to clarify, since this has been provided for the other steps and here it merely says that the samples are washed in RNA-editing buffer and then incubated at 37C overnight, which could be confusing.

We appreciate the close examination of the method section. The Method section ("*in situ* RNA editing and RNA-seq library preparation") has been modified with clarifications of the amount of RNA-editing buffer.

8. For the PFA-fixed samples – could the authors please comment on how the fixation could affect the application of their method together with long-read sequencing, since I would expect the de-cross-linking step to only be partial and the remaining cross-links could interfere with the RT step.

Indeed, PFA fixation could affect the long-read sequencing workflow, since it could lead to truncated RT libraries due to both RNA fragmentation during reverse crosslinking and as the reviewer mentioned, the incomplete de-crosslinking interfering with the readthrough of RT enzymes. We suggest users of INSCRIBE adopt the MeOH fixation workflow if they intend to perform long-read sequencing.

Responses to Reviewer #3:

Authors report a novel technique to identify and study RNA-RNA binding protein interactions in cell culture and tissues. The so called IN situ Sensitive Capture of RNA-protein Interactions in Biological Environments (INSCRIBE) method uses APOBEC1-nanobody to bind to RBPs. As a proof-of-principle, authors were able to capture and identify two RBPs - RBFOX2 and TDP-43 in cells and tissue slices. The methodology used by authors is appropriate and the results obtained support their hypothesis and claims. INSCRIBE is an innovative technology which has good potential to study RNA-RBP interactions in various samples and is a better method compared to existing techniques with application in biomedical research. The manuscript can be accepted for publication after authors address the comments below;

-What was the rationale for selecting RBFOX2 and TDP-43 proteins for detection? There are several RBPs which carry binding motifs with different lengths. Both RBFOX2 and TDP-43 are able to recognize nucleotide sequence with six bases. What about RBPs that recognize longer nucleotide sequences? Wondering about the general applicability of this method to a wide range of mammalian RBPs

We thank the reviewer for bringing these up. We chose RBFOX2 and TDP-43 because (1) they are both well-studied RBPs with well-defined binding motifs; (2) RBFOX2 in particular, is a good RBP to benchmark INSCRIBE performance since our lab has extensive data on RBFOX2 STAMP and eCLIP.

While “UGCAUG” and “UGUGUG” are the canonical binding motif for RBFOX2 and TDP-43 respectively, the two RBPs also recognize alternative motifs of the same or different lengths. In the *de novo* motif discovery software HOMER, the output motif length can be defined by the user. We used the length of six bases, which is a balance to get motif specificity while lowering the computational cost (a longer motif would take more time to compute). In light of this, INSCRIBE can surely discover motifs longer than 6nt. For example, when we set the motif length of 8nt for TDP-43 (known to bind GU repeats), the output top motif is “GUGUGUGU” (as shown in the figure below).

-What should be the ideal purity range of APOBEC1-nanobody to obtain consistent results

We have purified 2 batches of APOBEC1-nanobody so far and they are both >95% purity estimated by a coomassie-stained SDS-PAGE gel. Though we haven't thoroughly examined the batch effect of the enzyme yet, both batches had similar performance in the INSCRIBE experiment in our experience. Therefore, I would suggest the protein purity to be >95% and it has now been mentioned in the manuscript.

-Authors propose using this method to study RNA-RBP interactions in clinical/patient samples although no data is provided. Is it possible to use this technique to study RNA-RBP interactions in plasma samples? One important reference is missing (Cell Genom . 2023 Apr 20;3(5):100303. doi: 10.1016/j.xgen.2023.100303. eCollection 2023 May 10.)

We thank the reviewer for bringing this to our attention. At this stage, we have not applied INSCRIBE in clinical or patient samples yet, but in principle it could be straightforward since we have successfully profiled RNA-RBP interactions with PFA-fixed mouse brain tissue. However, INSCRIBE is not suitable for plasma samples because the experiments are generally set up on a slide which anchors the fixated tissue. RNA-RBP interaction in plasma samples or biofluids is beyond the scope of our study.

-Figure 6: Not clear on how many brain slices were prepared per mouse and how many mice were used in the study? Why did the authors use female mice? These details should be clarified in the manuscript

The mouse tissue experiments were a proof of concept that INSCRIBE could work with fixed tissue samples. Therefore, we did not select sex or age on purpose. Two mice were used in this study, one for the methanol fixation and the other for the formaldehyde fixation. Both were 2-month-old female mice. Each individual replicate of either experimental or control groups was prepared from one single slice of the brain (20µm coronal section). These details have been added to the revised Method “*in situ* RNA editing and RNA-seq library preparation” and “Animal” section.

-Minor comments: Author should use proper format while typing references in text. eg - page 3, paragraph 1, line 45 - "events1-3." should be "events.1-3". Punctuation "." should come before the reference numbers. Need to check for this throughout the manuscript

There are multiple formatting rules that different journals follow. Springer did not specify the ordering rule of punctuation and reference number. Through studying multiple examples that were published on Nature Communication, all of them have reference numbers before punctuation. Therefore, we did not change our previous formatting.

References section: Few references do not have proper journal format eg: 19, 22, 25 - words starting with upper case letters Vol and page numbers are missing for some listed references

We have made sure the references in the revised manuscript are properly cited.

References

- Begg, B. E., Jens, M., Wang, P. Y., Minor, C. M., & Burge, C. B. (2020). Concentration-dependent splicing is enabled by Rbfox motifs of intermediate affinity. *Nature Structural & Molecular Biology*, 27(10), 901–912.
- Dominguez, D., Freese, P., Alexis, M. S., Su, A., Hochman, M., Palden, T., Bazile, C., Lambert, N. J., Van Nostrand, E. L., Pratt, G. A., Yeo, G. W., Graveley, B. R., & Burge, C. B. (2018). Sequence, Structure, and Context Preferences of Human RNA Binding Proteins. *Molecular Cell*, 70(5), 854–867.e9.
- Feng, H., Bao, S., Rahman, M. A., Weyn-Vanhentenryck, S. M., Khan, A., Wong, J., Shah, A., Flynn, E. D., Krainer, A. R., & Zhang, C. (2019). Modeling RNA-Binding Protein Specificity In Vivo by Precisely Registering Protein-RNA Crosslink Sites. *Molecular Cell*, 74(6), 1189–1204.e6.
- Kuret, K., Amalietti, A. G., Jones, D. M., Capitanich, C., & Ule, J. (2022). Positional motif analysis reveals the extent of specificity of protein-RNA interactions observed by CLIP. *Genome Biology*, 23(1), 191.
- Yee, B. A., Pratt, G. A., Graveley, B. R., Van Nostrand, E. L., & Yeo, G. W. (2019). RBP-Maps enables robust generation of splicing regulatory maps. *RNA*, 25(2), 193–204.
- Kofman, E., Yee, B., Medina-Munoz, H.C. et al. FLARE: a fast and flexible workflow for identifying RNA editing foci. *BMC Bioinformatics* 24, 370 (2023).
- Lorenz, D.A., Her, H.L., Shen, K.A. et al. Multiplexed transcriptome discovery of RNA-binding protein binding sites by antibody-barcode eCLIP. *Nat Methods* 20, 65–69 (2023).
- Brannan, K.W., Chaim, I.A., Marina, R.J. et al. Robust single-cell discovery of RNA targets of RNA-binding proteins and ribosomes. *Nat Methods* 18, 507–519 (2021).
- Medina-Munoz, H.C., Kofman, E., Jagannatha, P. et al. Expanded palette of RNA base editors for comprehensive RBP-RNA interactome studies. *Nat Commun* 15, 875 (2024).

Reviewers' Comments:

Reviewer #1:

Remarks to the Author:

I would like to thank the authors for carefully considering my points and largely addressing them. In particular, I think the new, extended comparisons between INSCRIBE and eCLIP (e.g. Supplementary Figure 3) and the PacBio analysis strengthen the manuscript.

Comments

The difference in regional selection (Supplementary Figure 3C, intron v 3' UTR) and the exploration of the different motifs and spatial resolutions between eCLIP and INSCRIBE are important factors that it will be important for adopters of the method to understand. I think this is well explained in the new section.

I find the metaprofile-style RNA map plots (e.g. from the authors' lab (Yee et al. 2019, Fig. 2D and Fig. 3 for RBFOX2) more intuitive to understand regulatory principles and would have preferred to see these plotted, but perhaps it was not possible to derive a meaningful plot from the lower coverage data. However, from the summarised bar-plot maps in the rebuttal document, I agree with the authors' conclusions that the resolution of INSCRIBE does not really allow exploration of splicing regulatory information (e.g. RBFOX2 eCLIP HEK (in) shows enrichment in the upstream intron, but RBFOX2 INSCRIBE 100M (in) shows it in the downstream exon and at half the odds ratio; RBFOX2 eCLIP HEK (ex) and RBFOX2 INSCRIBE 100M (ex) show slightly better concordance).

I note the authors general point that in-depth splicing regulatory pattern discovery is beyond the scope of their manuscript, and I concur. However, I do think knowing whether it is possible to recover already known regulatory patterns from RBFOX2 eCLIP data using INSCRIBE is something a reader or adopter of the method would want to know as a current limitation of the method. Hence I would suggest that it would be helpful to include these conclusions in the main manuscript as a supplementary figure panel.

Reviewer #2:

Remarks to the Author:

The authors have satisfactorily answered all my comments and the additional data they generated supports the validity of their method.

Reviewer #3:

Remarks to the Author:

The authors have revised the manuscript based on previous comments. Their response to concerns, criticism and questions on methodology is nicely articulated in their rebuttal letter, and in the revised manuscript.

The INSCRIBE method proposed will advance the field by providing additional tools to study RNA-RBP interactions. There are limitations in this work which the authors are know and are planning to address them in their future work. Publishing this work would certainly encourage other scientists in the field to refine or perhaps develop better methods to study RNA-RBP interactions especially when testing low quantity samples or in patient samples.

We thank the reviewers for their careful reading of the manuscript. We addressed the additional reviewer's comments below in blue text.

Reviewer #1 (Remarks to the Author):

I would like to thank the authors for carefully considering my points and largely addressing them. In particular, I think the new, extended comparisons between INSCRIBE and eCLIP (e.g. Supplementary Figure 3) and the PacBio analysis strengthen the manuscript.

Comments

The difference in regional selection (Supplementary Figure 3C, intron v 3' UTR) and the exploration of the different motifs and spatial resolutions between eCLIP and INSCRIBE are important factors that it will be important for adopters of the method to understand. I think this is well explained in the new section.

I find the metaprofile-style RNA map plots (e.g. from the authors' lab (Yee et al. 2019, Fig. 2D and Fig. 3 for RBFOX2) more intuitive to understand regulatory principles and would have preferred to see these plotted, but perhaps it was not possible to derive a meaningful plot from the lower coverage data. However, from the summarised bar-plot maps in the rebuttal document, I agree with the authors' conclusions that the resolution of INSCRIBE does not really allow exploration of splicing regulatory information (e.g. RBFOX2 eCLIP HEK (in) shows enrichment in the upstream intron, but RBFOX2 INSCRIBE 100M (in) shows it in the downstream exon and at half the odds ratio; RBFOX2 eCLIP HEK (ex) and RBFOX2 INSCRIBE 100M (ex) show slightly better concordance).

I note the authors general point that in-depth splicing regulatory pattern discovery is beyond the scope of their manuscript, and I concur. However, I do think knowing whether it is possible to recover already known regulatory patterns from RBFOX2 eCLIP data using INSCRIBE is something a reader or adopter of the method would want to know as a current limitation of the method. Hence I would suggest that it would be helpful to include these conclusions in the main manuscript as a supplementary figure panel.

We agree with the reviewer that it'd be helpful for the audience to learn about INSCRIBE's performance in discovering regulatory patterns. We have added these figures as Supplementary Figure 5E and explained the limitations of INSCRIBE in discovering regulatory patterns at line 278-284.

Reviewer #2 (Remarks to the Author):

The authors have satisfactorily answered all my comments and the additional data they generated supports the validity of their method.

Reviewer #3 (Remarks to the Author):

The authors have revised the manuscript based on previous comments. Their response to concerns, criticism and questions on methodology is nicely articulated in their rebuttal letter, and in the revised manuscript.

The INSCRIBE method proposed will advance the field by providing additional tools to study RNA-RBP interactions. There are limitations in this work which the authors are know and are planning to address them in their future work. Publishing this work would certainly encourage other scientists in the field to refine or perhaps develop better methods to study RNA-RBP interactions especially when testing low quantity samples or in patient samples.